# vATTENTION: VERIFIED SPARSE ATTENTION

**Aditya Desai**[α*]   **Kumar Krishna Agrawal**[α*]   **Shuo Yang**[α†]   **Alejandro Cuadron**[α, β]
**Luis Gaspar Schroeder**[α]   **Matei Zaharia**[α]   **Joseph E. Gonzalez**[α]   **Ion Stoica**[α]
[α] Electrical Engineering and Computer Sciences, University of California, Berkeley,
[β] Department of Computer Sciences, ETH Zürich

## ABSTRACT

State-of-the-art sparse attention methods for reducing decoding latency fall into two main categories: approximate top-$k$ (and its extension, top-$p$) and recently introduced sampling-based estimation. However, these approaches are fundamentally limited in their ability to approximate full attention: they fail to provide consistent approximations across heads and query vectors and, most critically, lack guarantees on approximation quality, limiting their practical deployment. We observe that top-$k$ and random sampling are complementary: top-$k$ performs well when attention scores are dominated by a few tokens, whereas random sampling provides better estimates when attention scores are relatively uniform. Building on this insight and leveraging the statistical guarantees of sampling, we introduce vAttention, the first practical sparse attention mechanism with user-specified $(\epsilon, \delta)$ guarantees on approximation accuracy (thus, "verified"). These guarantees make vAttention a compelling step toward practical, reliable deployment of sparse attention at scale. By unifying top-k and sampling, vAttention outperforms both individually, delivering a superior quality–efficiency trade-off. Our experiments show that vAttention significantly improves the quality of sparse attention (e.g., $\sim$4.5 percentage points for Llama-3.1-8B-Inst and Deepseek-R1-Distill-Llama-8B on RULER-HARD ), and effectively bridges the gap between full and sparse attention (e.g., across datasets, it matches full model quality with upto 20x sparsity). We also demonstrate that it can be deployed in reasoning scenarios to achieve fast decoding without compromising model quality (e.g., vAttention achieves full model quality on AIME2024 at 10x sparsity with up to 32K token generations).

## 1 INTRODUCTION

As the application of AI expands and workflows grow more complex, the volume of context that large language models (LLMs) must maintain is rapidly increasing(Touvron et al., 2023; Achiam et al., 2023; Liu et al., 2024a). However, Scaled Dot Product Attention (SDPA) operator, the core operation behind the success of transformer architectures(Vaswani, 2017; Brown et al., 2020), is not well suited for handling such long contexts during generation. Large contexts produce massive key–value (KV) embedding caches, and in autoregressive models, these caches must be repeatedly read for every new token prediction. This makes the decoding step inherently memory-bound and time-consuming (Kim et al., 2023). The problem becomes even more severe when the KV caches exceed available GPU memory and must be offloaded to CPU RAM, requiring costly transfers across the CPU–GPU boundary. These bottlenecks highlight a fundamental scalability issue in attention mechanisms, limiting the ability of LLMs to efficiently consider long contexts. A mitigation strategy is sparse attention, which reduces memory movement by attending only to a subset of tokens in the KV cache. A good sparse attention would offer highly accurate approximations of full attention that it replaces.

The core abstract problem in approximating Scaled Dot Product Attention (SDPA; see Eq. 3) is estimating the sum of $n$ quantities (scalars for denominator and vectors for numerator in SDPA). Since, in hindsight, the tokens that contribute most to the attention output are those with the highest $a_i||\mathbf{v}_i||_2$ (Desai et al., 2025), a natural approach is to choose tokens $i$ with the largest query–key inner products $k_i^\top q$, also known as the top-$k$ approach (top-$p$, its extension, chooses budgets per

---

*Equal contribution, † system lead , correspondence:{`apdesai`,`kagrawal`,`istoica`}@`berkeley.edu`

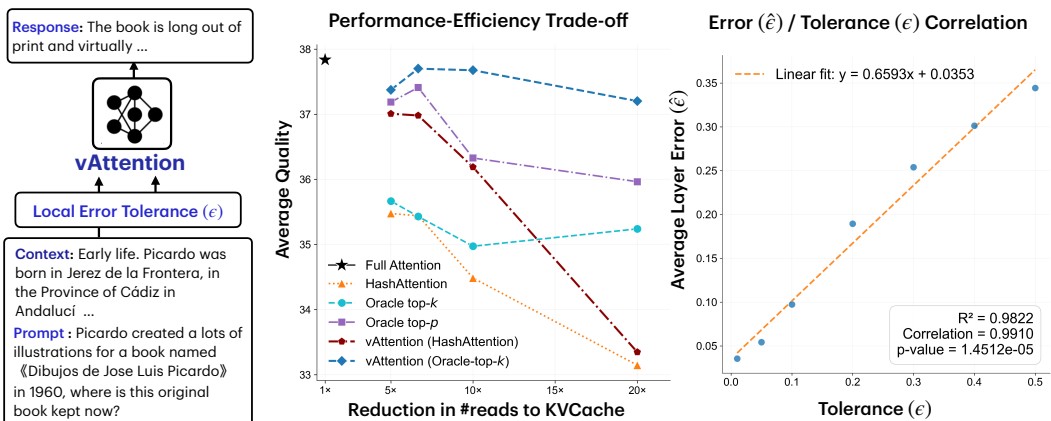

Figure 1: [**Left**:] vAttention accepts user tolerance parameter $\epsilon$ and ensures that sparse attention errors are controlled to be within this tolerance. [**Middle**] vAttention achieves a SOTA trade-off, outperforming leading methods like HashAttention and even a strong oracle top-$p$ on a mix from long-context benchmarks (RULER32K, LongBench, Loogle). [**Right**] There is a strong correlation between the approximation error in the layer attention output and the user-defined parameter $\epsilon$ accepted by vAttention with verified denominator-only approximation, validating the practical relevance of $\epsilon$ parameter

head). Thus, sparse attention research is dominated by approaches designed to approximate top-$k$ (Xiao et al., 2024; Tang et al., 2024; Desai et al., 2025; Li et al., 2024b; Hooper et al., 2024; Zhang et al., 2025) and top-p (Zhu et al., 2025) efficiently. However, any deterministic sparsity, such as top-$k$, assumes that the contextual embedding of the current token can be determined by a small set of specific tokens in the context as if information is encoded in discrete units – an assumption that is clearly not true in input layer (e.g. next word does not depend on only a few words ) and cannot be reliably assumed in deeper layers. It is not surprising that attention scores are not always sharply distributed, and in such cases, top-$k$ methods fail to give a good approximation (see Fig. 2). While top-$k$ tokens often dominate attention outputs, contextual meaning arises from the entire distribution of key–value vectors across all tokens. This motivates the statistical perspective behind Verified Sparse Attention (vAttention).

Recently, LSH-based sampling was used to approximate attention (Chen et al., 2024). vAttention is inspired by this work. vAttention is based on a key observation for approximating a sum of $n$ terms. If the sum is sharply dominated by a few terms, selecting those terms provides the best approximation. Conversely, if the terms are similar in value, a case in which top-$p$ leads to choosing an unnecessarily large number of terms, a sampling-based estimator can approximate the sum with a small sample. vAttention combines both strategies, and adjusts the sample size to guarantee a user-specified $(\epsilon, \delta)$ approximation to the target sum. Using this intuition, vAttention approximates both the numerator and denominator to a specified accuracy, ensuring that the overall error in attention output incurs at most an $\epsilon$ relative error with probability $(1 - \delta)$, thus providing a "verified" sparse attention. To the best of our knowledge, vAttention is the first practical algorithm to provide approximation guarantees while giving users explicit control over the quality–efficiency tradeoff. This not only provides state-of-the-art quality on sparse attention, but it also makes a compelling argument in favor of deploying sparse attention reliably in the wild.

We extensively evaluate vAttention across diverse models (Llama-3.1-8B-Instruct, Deepseek-R1-Distill-Llama-8B, Mistral-7B-Instruct-v0.3) and benchmarks (RULER (Hsieh et al., 2024), Long-Bench (Bai et al., 2024), Loogle (Li et al., 2023), AIME (Maxwell-Jia, 2024)) and composing it with oracle-top-$k$, and HashAttention (Desai et al., 2025), a state-of-the-art approximate top-$k$. We find that vAttention consistently achieves higher accuracy than top-$k$ methods and delivers a superior quality–efficiency trade-off, surpassing even the strongest oracle top-p baseline (See Figure 1. For example, at 10% sparsity, vAttention combined with HashAttention improves RULER32K-HARD accuracy by upto 4.5 percentage points over HashAttention across models. Furthermore, owing to its low approximation error, vAttention supports accurate long-form generation, producing sequences of up to 32K tokens while matching full-attention accuracy on AIME(Maxwell-Jia, 2024). Additionally,

we show that there is a near-perfect correlation between the user-specified tolerance $\epsilon$ and the average empirically observed error in attention outputs, showcasing the effectiveness of vAttention in exposing the quality–efficiency trade-off of sparse attention to the end user (see Figure 1).

## 2 RELATED WORK

Existing work on sparse attention can be categorized into the following types, covering early explorations to recent efforts. Detailed related work is presented in Appendix B.

**Static Sparse Attention and KV Cache compression methods** Early sparse attention methods used fixed patterns to limit tokens during decoding. For instance, StreamingLLM (Xiao et al., 2023) attends to "attention sinks" (early tokens) and a sliding window of recent tokens. Later work (Zhang et al., 2023; Xiao et al., 2024) showed that such static patterns fail to generalize, motivating dynamic sparsity. StreamingLLM's key insight—that sinks and local windows are essential—remains central to subsequent methods. Another direction compresses the KV cache by discarding tokens, as in ScissorHands (Liu et al., 2024b), H2O (Zhang et al., 2023), FastGen (Ge et al., 2023), and SnapKV (Li et al., 2024a). While memory-efficient, these approaches lack generality, since irreversible pruning struggles in settings like multi-turn dialogue, where token relevance shifts across turns.

**Approximate top-$k$ based Sparse Attention** A class of sparse attention methods approximates top-$k$ token selection—identifying tokens with the highest query–key inner products—since exact computation is $O(nd)$ and undermines efficiency. Examples include Double Sparsity (Yang et al., 2015), which sparsifies partial channels; Loki (Singhania et al., 2024), which applies low-rank decomposition; and InfLLM (Xiao et al., 2024) and Quest (Tang et al., 2024), which use page-level approximations. As top-$k$ identification is essentially an inner product search problem (Desai et al., 2025), many methods adapt approximate nearest neighbor (ANN) techniques: PQCache (Zhang et al., 2025) leverages product quantization, SqueezeAttention (Hooper et al., 2024) employs hierarchical clustering, Retrieval Attention (Li et al., 2024b) adopts graph-based ANN search, and HashAttention (Desai et al., 2025) encodes queries and keys as bit signatures. These approaches improve scalability by narrowing the search to promising tokens, but their dependence on oracle top-$k$ selection imposes a fundamental limitation. As demonstrated in MagicPig (Chen et al., 2024) and further analyzed here, even access to the exact top-$k$ tokens under full attention does not always suffice to approximate the original output, highlighting the need to move beyond top-$k$ selection in sparse attention design.

**Approximate Top-P based Sparse Attention** A key limitation of top-$k$–based sparse attention is that a fixed sparsity level fails to generalize across modules. To address this, recent work adopts Top-p coverage, selecting a variable number of tokens whose cumulative attention scores exceed a threshold $p$, thereby adapting to varying importance distributions while offering error control. Exact Top-p, however, is even costlier than top-$k$ as it requires sorting or aggregating all scores; methods therefore approximate coverage—for example, Tactic (Zhu et al., 2025) models attention decay with a power-law distribution to estimate the required number of tokens. As we show, Top-p is not the most efficient way to achieve a target error bound; more principled mechanisms, including vAttention, attain comparable or better accuracy with fewer tokens.

**MagicPig: LSH sampling-based Sparse Attention** MagicPig (Chen et al., 2024) was among the first to highlight the issues with top-$k$–based sparse attention. It employs Locality Sensitive Hashing (LSH)(Gionis et al., 1999) to select tokens for attention computation. While LSH is suboptimal for approximate nearest neighbor (ANN) search due to its data-agnostic projections, in this context, it provides a principled sampling-based mechanism for approximating attention (Luo & Shrivastava, 2018). Tokens retrieved via LSH have associated probabilities reflecting how they were sampled under the randomized construction of the LSH table. Early exploration of vAttention was inspired by MagicPig, and we elaborate further on the attention computation in subsequent sections. Another related work, SampleAttention (Zhu et al., 2024), samples structured patterns to approximate attention.

**Theoretical exploration of top-k and sampling** There is extensive literature on theoretically analyzing softmax attention and introducing approximations for faster training and inference, including Performers Choromanski et al. (2020), Linformer Wang et al. (2020), HyperAttention Han et al. (2023), and KNN-Attention Haris (2024). These works develop subquadratic attention mechanisms that must be trained end-to-end and are primarily focused on performance (training and inference) post training from scratch. In contrast, our work targets inference-time attention approximation to

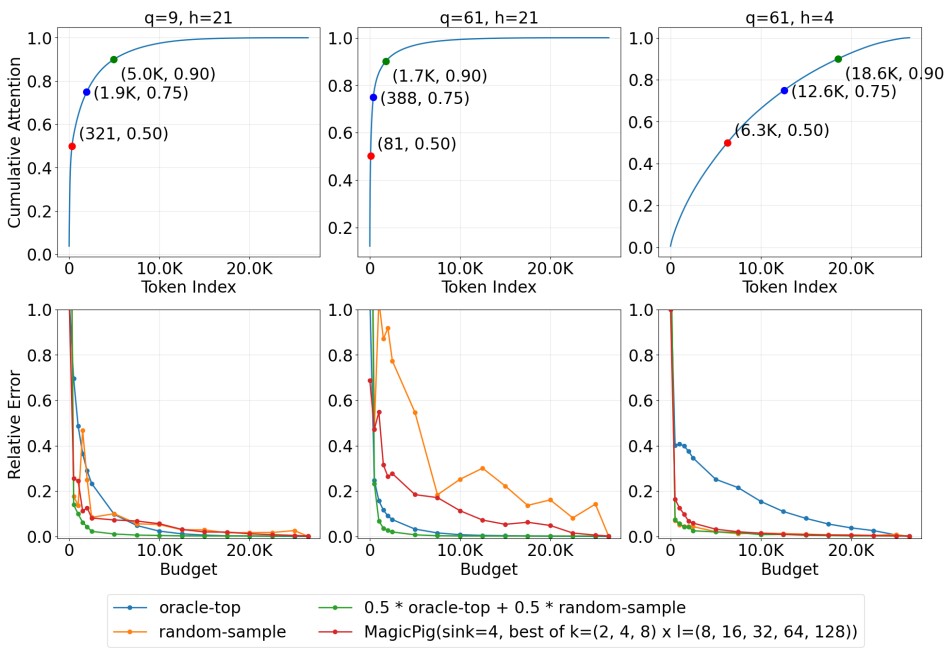

Figure 2: Top pane: cumulative sum of attention scores sorted in descending order of magnitude, showing the number of tokens required to achieve a $p \in (0, 1)$ coverage over the scores. Bottom: relative local attention errors across token budgets, indexed by head $h$ and query $q$ index

accelerate decoding in off-the-shelf models. A key point of comparison is with KNN-Attention Haris (2024), which investigates top-k selection and sampling through lazy Gumbel sampling and proposes top-k plus sampling as a relaxation. However, their emphasis is on training, with theoretical treatment and objectives that differ substantially and do not address inference acceleration for existing models. vAttention, by contrast, provides a simple, practical theoretical foundation that directly yields an implementable algorithm, which we rigorously evaluate across multiple off-the-shelf models and benchmarks.

Existing methods for inference time acceleration lack guarantees on the quality of approximation. In contrast, vAttention offers explicit control over approximation quality, providing both reliability and flexibility.

## 3 BACKGROUND AND MOTIVATION

We begin by describing the three categories of attention computation considered in this paper. For clarity, we restrict our exposition to the case of batch size one with a single query vector. Consider KVCache $K, V : n \times d$ and query vector $q : 1 \times d$.

**Full Scaled Dot Product Attention (SDPA)**

$$\text{SDPA}(\mathbf{K}, \mathbf{V}, \mathbf{q}) = \sum_{i=1}^{n} (a_i \mathbf{V}[i]) \text{ where } a_i = \frac{\exp \langle \mathbf{K}[i], q \rangle}{\sum_{j=1}^{n} \exp \langle \mathbf{K}[j], q \rangle} \tag{1}$$

where $a_i$ are referred to as attention scores. This represents full attention computation.

**Sparse Attention with deterministic index selection:** Let $S$ denote the sequence of indices selected by a deterministic method—such as attention sinks, sliding window, top-$k$ selection, or a combination of them. The sparse attention computation based on this deterministic index set is given by:

$$\text{SDPA}_S(\mathbf{K}, \mathbf{V}, \mathbf{q}) = \sum_{i \in S} (\hat{a}_i \mathbf{V}[i]) \text{ where } \hat{a}_i = \frac{\exp \langle \mathbf{K}[i], q \rangle}{\sum_{j \in S} \exp \langle \mathbf{K}[j], q \rangle} \tag{2}$$

**Sparse Attention with randomized index selection:** Let $S$ denote the sequence of indices selected by a randomized method—such as random sampling or MagicPig, and let $P$ be the corresponding

sequence of selection probabilities. Given $S$ and $P$, the attention computation is defined as:

$$\text{SDPA}_{S,P}(\mathbf{K}, \mathbf{V}, \mathbf{q}) = \sum_{(i,p_i)\in(S,P)} (\hat{a}_i \mathbf{V}[i]) \text{ where } \hat{a}_i = \frac{\frac{1}{p_i}\exp\langle\mathbf{K}[i], q\rangle}{\sum_{(j,p_j)\in(S,P)}\frac{1}{p_j}\exp\langle\mathbf{K}[j], q\rangle} \quad (3)$$

This definition subsumes deterministic index selection ( where probabilities associated are 1) and can be used to represent a selection that is a combination of deterministic and randomized selection.

We motivate our approach through a simple ablation. We use the following baseline: **oracle-top:** selects the tokens with the highest inner products, constrained by the budget, and uses deterministic attention computation. **random-sample:** uniformly samples a subset of tokens (without replacement) of size equal to the budget, and applies sampling-based attention computation. **MagicPig:** Does LSH-based index retrieval followed by sampling estimation. If more tokens are retrieved than the budget allows, a subset of size equal to the budget is randomly selected; otherwise, all retrieved tokens are used. We plot the configuration among ($k = \{2, 4, 8\} \times L = \{8, 16, 32, 64, 128\}$) that yields the best errors for a particular sparsity. We use a sample from the GSM-Infinite (Zhou et al., 2025) dataset of length $25K$ and use the last $128$ queries for attention computation.

As shown in Figure 2, the quality–efficiency tradeoff depends on the distribution of attention scores, and no baseline is universally superior. We observe that the ordering is inconsistent—for a given query across heads, or for a given head across queries—highlighting the need for dynamic behavior per head per query. Moreover, when attention scores are sharply distributed, oracle top-$k$ provides a better tradeoff. In contrast, random sampling performs better in the presence of a long tail (looking at the distribution of attention scores sorted in descending order). MagicPig, though based on importance sampling, also fails to outperform other methods consistently. Drawing inspiration from these observations, we propose to combine top-$k$ and sampling methods. As a representative, we use **oracle-top + random-sample:** using half the budget for oracle-top and the other half for sample. We find that this combination consistently yields superior results in all three cases. This hybrid strategy serves as a simplification of our proposed vAttention method.

## 4 VATTENTION: VERIFIED SPARSE ATTENTION

Restricting attention to only a fraction of tokens alters model behavior by introducing errors in the intermediate calculations of the numerator and denominator. These errors propagate, impacting per-head attention and, ultimately, the overall attention computation at each layer. While directly controlling the resulting errors in per-layer outputs or overall model behavior is mathematically challenging, we can effectively regulate the errors in these fundamental computations, which correlate strongly with per-layer attention deviations and, by extension, model behavior. vAttention provides recipe for $(\epsilon, \delta)$ verified computation for numerator, denominator and per-head attention. In general, $(\epsilon, \delta)$ verified-$\mathcal{X}$ algorithm is,

**Definition 4.1** (($\epsilon, \delta$)-verified-$\mathcal{X}$). *For any given computation $\mathcal{X} : R^{d_1} \to R^{d_2}$ for some $d_1, d_2 \in \mathbf{N}$, an algorithm $\mathcal{X}'$ is $(\epsilon, \delta)$-verified-$\mathcal{X}$ if the following holds for any $x \in R^{d_1}$*

$$\mathbf{Pr}\left(\frac{||\mathcal{X}'(x) - \mathcal{X}(x)||_2}{||\mathcal{X}(x)||_2} > \epsilon\right) < 1 - \delta \quad (4)$$

We will first describe the recipes for verified$-\mathcal{N}$ and verified$-\mathcal{D}$ for numerator and denominator computations. Then we show how to combine then for per-head attention.

### 4.1 VERIFIED-$\mathcal{D}$ AND VERIFIED-$\mathcal{N}$

Let the KV cache for a given attention head be denoted by $K, V \in \mathbb{R}^{n \times d}$ and the query by $q \in \mathbb{R}^{1 \times d}$. Both numerator and denominator are sum of $n$ terms. vAttention breaks down the problem of approximation into two parts: (1) identifying outlier or heavy-hitter tokens, and (2) approximating the residual long tail of tokens with similar attention scores using uniform random sampling. The key idea is two fold. First, if the heavy tokens are correctly identified, the contribution of the residual tail can be approximated with a small sample. Second, the convergence properties of uniform sampling can be leveraged to provide guarantees on approximation errors. We describe the exact algorithm and its mathematical foundations below.

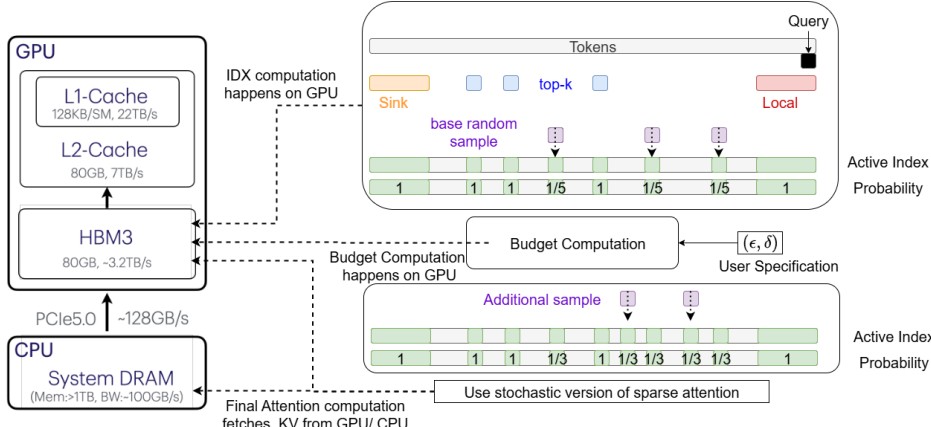

Figure 3: vAttention composes, sink, sliding window, and approximate top-k based attention along with random sampling based selection whose budget is governed by an adaptive sampling module which ensures user specified $(\epsilon, \delta)$ guarantees hold for each attention head every layer. The index computation and budget computation occur entirely on the GPU, and the final attention computation can retrieve the KV cache from either the GPU/CPU, depending on its location.

The index selection procedure in vAttention is illustrated in Figure 3. vAttention selects a mixture of deterministic and stochastic indices, and is parameterized by $f_s, f_l, f_t, f_b, \epsilon, \delta \in (0,1)$. For the deterministic component, vAttention includes sink indices $\mathcal{I}_s, |\mathcal{I}_s| = f_s n$, local window indices $\mathcal{I}_l, |\mathcal{I}_l| = f_l n$, and predicted top-k tokens $\mathcal{I}_t, |\mathcal{I}_t| = f_t n$. These correspond to the most salient tokens expected to dominate the attention distribution, i.e., those with high attention scores. vAttention can be composed with any off-the-shelf approximate top-k method. Incorporating such "heavy hitters" is crucial for strong performance, since the stochastic component—uniformly sampled indices—can only approximate the residual attention accurately with small sample when the remaining distribution does not have significant outliers (i.e., when residual attention scores have comparable magnitudes). Let $\mathcal{I}_f = \mathcal{I}_s \cup \mathcal{I}_l \cup \mathcal{I}_t$ and $|\mathcal{I}_s| + |\mathcal{I}_l| + |\mathcal{I}_t| = n_f$.

Let $n_s = n - n_f$ denote the number of residual indices. vAttention uniformly samples indices $\mathcal{I}_{dyn}$ from residual indices. The set of indices and associated sampling probabilities can be expressed as,

$$S = \mathcal{I}_f \cup \mathcal{I}_{dyn} \qquad P_i = \frac{|\mathcal{I}_{dyn}|}{n_s} \text{ if } i \in \mathcal{I}_{dyn} \text{ and } 1 \text{ otherwise} \tag{5}$$

Then the vAttention computation for numerator and denominator can be written as,

$$N = N_f + N_{dyn} = \sum_{i \in \mathcal{I}_f} \left(\exp \langle K[i], q \rangle V[i]\right) + \frac{n_s}{|\mathcal{I}_{dyn}|} \sum_{j \in \mathcal{I}_{dyn}} \left(\exp \langle K[j], q \rangle V[j]\right) \tag{6}$$

$$D = D_f + D_{dyn} = \sum_{i \in \mathcal{I}_f} \left(\exp \langle K[i], q \rangle\right) + \frac{n_s}{|\mathcal{I}_{dyn}|} \sum_{j \in \mathcal{I}_{dyn}} \left(\exp \langle K[j], q \rangle\right) \tag{7}$$

where $N_f$ (alt. $D_f$) comes from deterministic indices and $N_{dyn}$ (alt. $D_{dyn}$) comes from stochastic indices that estimate the rest of the numerator (alt. denominator).

The theoretical guarantees in vAttention arise from the careful choice of the sample size, i.e., $|\mathcal{I}_{dyn}|$. The sample size is selected to ensure that the attention approximation is $\epsilon$-relative accurate with probability $1 - \delta$. We now explain how the budget is chosen. The choice of sample size is guided by the following result on estimating the sum of $n$ quantities (See Appendix D for proof)

**Lemma 4.1 (Estimating vector sum).** *Let $\mathbf{s} = \sum_{i=1}^{n_s} \mathbf{r}_i, \mathbf{s} \in R^d$ be a sum of $n_s$ vector quantities $\mathbf{r}_i \in R^d \forall i$ which has to be estimated using a sample $\mathcal{I}_b$ of size $b$. Let $\Sigma$ be the covariance matrix for the population $\{\mathbf{r}_i\}_{i=1}^{n_s}$. Let $\hat{\mathbf{s}}_b = \frac{n_s}{b} \left(\sum_{i \in \mathcal{I}_b} \mathbf{r}_i\right)$ be the estimate. Let $\Phi$ be the CDF for the normal distribution. Then for a large enough $b$ if,*

$$b \geq \left(\Phi^{-1}\left(1 - \frac{\delta}{2}\right) \frac{n_s \sqrt{\mathbf{Tr}(\Sigma)}}{\tau}\right)^2 \quad then \quad \mathbf{Pr}(||\hat{\mathbf{s}} - \mathbf{s}||_2 > \tau) \leq \delta \tag{8}$$

*for any arbitrary $\tau \in R$ and $\delta \in (0,1)$.*

**Algorithm 1** vAttention

**Require:** $\mathcal{X} \in \{\mathcal{N}, \mathcal{D}, \text{SPDA}\}$:computation to keep verified, KVCache $\mathbf{K}, \mathbf{V} : n \times d$, $q : 1 \times d$. Parameters $f_s, f_l, f_t \in (0,1)$ : fraction of total tokens for sink, sliding window and top-k tokens. Adaptive sampling parameters $f_b \in (0,1)$ : base sampling rate, $\varepsilon, \delta \in (0,1)$ user specified guarantee.

1: $\mathcal{I}_s \leftarrow \{0, 1, \ldots, \lfloor f_s n \rfloor - 1\}$
2: $\mathcal{I}_l \leftarrow \{n - \lfloor f_l n \rfloor, \ldots, n-1\}$
3: $\mathcal{I}_t \leftarrow \texttt{pred-top-index}(f_t, n, K, V, q)$
4: $\mathcal{I}_f \leftarrow \mathcal{I}_s \cup \mathcal{I}_l \cup \mathcal{I}_t$ , $\mathbf{p}_f = \mathbf{1}^{|\mathcal{I}_f|}$
5: $n_s \leftarrow n - |\mathcal{I}_f|$
6: $b \leftarrow \texttt{budget}_{\mathcal{X}}(\mathcal{I}_f, f_b, n_s, \varepsilon, \delta, \mathbf{K}, \mathbf{V}, q)$
7: $\mathcal{I}_{\text{dyn}} \leftarrow \texttt{uniform-sample}(\mathcal{I}_f, b, n)$
8: $\mathbf{p}_{\text{dyn}} \leftarrow \frac{b}{n_s} \cdot \mathbf{1}^{|\mathcal{I}_{\text{dynamic}}|}$
9: $S \leftarrow [\mathcal{I}_f, \mathcal{I}_{\text{dynamic}}]$
10: $p \leftarrow [\mathbf{p}_f, \mathbf{p}_{\text{dynamic}}]$
11: **return** $\text{SDPA}_{S,p}(\mathbf{K}, \mathbf{V}, q)$

**Algorithm 2** $\texttt{budget}_{\mathcal{X}}$

**Require:** KVCache $\mathbf{K}, \mathbf{V} : n \times d$, $q : 1 \times d$ , Parameters: $n_s$ sampling range, $f_b \in (0,1)$ : base sampling rate, $\epsilon, \delta \in (0,1)$ probabilistic guarantee required by user, $\mathcal{I}_f$ static index.

1: $\mathcal{I}_{bs} \leftarrow \texttt{uniform-sample}(\mathcal{I}_f, f_b, n)$
2: **if** $\mathcal{X} = \mathcal{D}$ **then**
3: $\quad \hat{\sigma^2}, \hat{D} \leftarrow \texttt{get-stats}(\mathcal{I}_f, \mathcal{I}_{bs}, K, V, q)$
4: $\quad b = b_{\mathcal{D}}(\epsilon, \delta, \hat{\sigma}, \hat{D}, K, V, q)$
5: **else if** $\mathcal{X} = \mathcal{N}$ **then**
6: $\quad Tr(\hat{\Sigma}), ||\hat{N}||_2^2 \leftarrow \texttt{get-stats}(\mathcal{I}_f, \mathcal{I}_{bs}, K, V, q)$
7: $\quad b = b_{\mathcal{N}}(\epsilon, \delta, \hat{\mathbf{Tr}}(\Sigma), ||\hat{N}||_2, K, V, q)$
8: **else**
9: $\quad Tr(\hat{\Sigma}), \hat{\sigma^2}, ||\hat{N}||_2^2, \hat{D} \leftarrow$
$\quad \texttt{compute-stats}(\mathcal{I}_f, \mathcal{I}_{bs}, K, V, q)$
10: $\quad b = b_{\mathbf{SDPA}}(\epsilon, \delta, \hat{\mathbf{Tr}}(\Sigma), \hat{\sigma}, ||\hat{N}||_2, \hat{D}, K, V, q)$
11: **end if**
12: **return** $b$

**Comment** We leverage the Central Limit Theorem (CLT) to approximate the sum using a sufficiently large sample. We can obtain a similar result for scalar quantities by setting $d = 1$ in the theorem above. Detailed empirical analysis of using optimistic CLT and conservative Hoeffding's method for budget computation is presented in E.

We can use the above lemma to compute budget required for $(\epsilon, \delta)$ approximations of the numerator and denominator independently as mentioned in Corollary D.2 D.3. These are obtained by setting $\tau = \epsilon ||N||_2$ and $\tau = \epsilon D$ in the numerator and denominator cases, respectively.

## 4.2 VERIFIED-SDPA

Let $b_D(\epsilon, \delta, \sigma, D, K, V, q)$ (resp. $b_N(\epsilon, \delta, \mathbf{Tr}(\Sigma), ||N||_2, K, V, q)$) denote the minimum budget required to achieve an $(\epsilon, \delta)$-approximation for the denominator (resp. numerator). When parameters are clear from the context, we will drop those from the expression for convenience. The individual approximation results for the numerator and denominator can be combined to yield a bound on the quality of the approximated attention output, as stated in the lemma below.

**Lemma 4.2.** *If $b_D$ and $b_N$ are chosen such that we have $(\epsilon_1, \delta_1)$ and $(\epsilon_2, \delta_2)$ approximation on numerator and denominator respectively and $\epsilon_2 < 0.5$, then using $b = \max(b_D, b_N)$ ensures that*

$$\mathbf{Pr}\left(\left\|\frac{N}{D} - \frac{\hat{N}}{\hat{D}}\right\|_2 > 2(\epsilon_1 + \epsilon_2)\left\|\frac{N}{D}\right\|_2\right) < (\delta_1 + \delta_2) \tag{9}$$

**Comment.** The lemma above shows that if both the numerator and denominator are well approximated, then the overall attention output is also well approximated.

The results above can be combined into a single theorem providing an algorithm to select budget for $(\epsilon, \delta)$ approximation of attention output.

**Theorem 4.3** ($(\epsilon, \delta)$ **verified**-SDPA$(\mathbf{K}, \mathbf{V}, \mathbf{q})$). *Let $\Sigma$ be the covariance matrix for the population $\{\exp \langle K[i], q V[i] \rangle\}_{i \in \bar{\mathcal{I}}_f}$. Let $\sigma$ be the standard deviation for the population $\{\exp \langle K[i], q \rangle\}_{i \in \bar{\mathcal{I}}_f}$. Then, if the budget is*

$$b \geq \min_{\epsilon' \in (0, \epsilon), \delta' \in (0, \delta)} \left[ \max\left( b_D\left(\frac{\epsilon'}{2}, \delta'\right), b_N\left(\frac{\epsilon - \epsilon'}{2}, \delta - \delta'\right) \right) \right] \quad then$$

$$\mathbf{Pr}(||vAttention(K, V, q) - SDPA(K, V, q)||_2 > \epsilon ||SDPA(K, q, v)||_2) \leq \delta \tag{10}$$

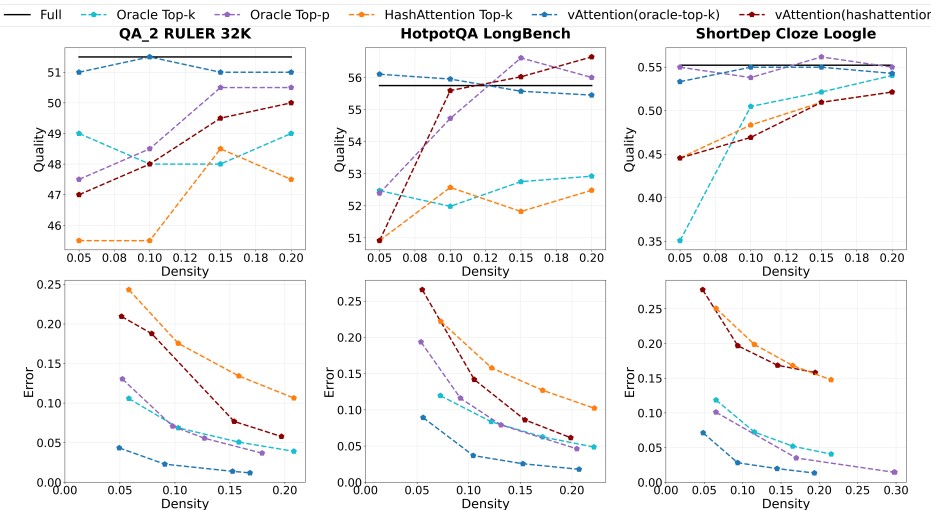

Figure 4: Pareto curves (Quality and Error vs. Density) for different baselines and their combination with vAttention across different datasets/benchmarks for Llama-3.1-8B-Instruct model. More pareto results are in Appendix A.1

Let $b_{\text{SDPA}}(\epsilon, \delta, \Sigma\sigma, ||N||_2, D, K, V, q)$ denote the minimum budget required for an $(\epsilon, \delta)$ approximation of SDPA attention. When parameters are clear from the context, we will drop those in the expression for convenience. In practice, $\Sigma$ and $\sigma$ as well as exact $||N||_2$ and $D$ are not known apriori. However, we can use a base sample from residual tokens to estimate them in order to compute these quantities. In our algorithm, vAttention is parameterized by $f_b$, which denotes the fraction of $n_s$ to be used as a sample for this estimation. The complete algorithms are provided in Algo. 1 and Algo. 2.

**Implementation Details** The index computation for vAttention is efficient even when the KV cache resides on the CPU. The computation is naturally vectorizable and well-suited for GPUs. Although vAttention requires partial access to the KV cache for budget calculations, this can be handled on the GPU using a small random cache that is incrementally populated and updated during token generation. When combined with approximate top-$k$ methods such as HashAttention or DoubleSparsity, the auxiliary structures (e.g., bit cache or label cache) for top-$k$ selection—orders of magnitude smaller than the full KV cache—can be stored directly on the GPU. For example, HashAttention requires only 32 bits per token per head for its bit cache.

## 5 EXPERIMENTS

Table 1: Average performance of different methods on RULER32K-HARD benchmark (consists of 7 datasets from RULER) at 10% sparsity. HashAttention is denoted as HAT. Detailed results are in Appendix A.3

|  | Llama-3.1-8B-Inst | Dpsk-R1-Distill-Llama-8B | Mistral-7B-Inst-v0.3 |
|---|---|---|---|
| SDPA | 88.74 | 65.41 | 64.05 |
| oracle-top-$k$ | 87.18 | 64.87 | **64.37** |
| vAttention(oracle-top-$k$) | **88.61** | **65.15** | 64.12 |
| HAT | 81.94 | 60.70 | 54.66 |
| vAttention(HAT) | **86.56** | **65.06** | **56.90** |

We perform an elaborate evaluation of vAttention against oracle and approximate baselines on multiple datasets, models, and generation lengths. The evaluation setup is explained below.

**Datasets and Models.** We evaluate vAttention on four benchmark suites: RULER (32K context length) Hsieh et al. (2024), LongBench Bai et al. (2024), and Loogle (truncated to 16K) Li et al. (2023), providing a broad basis for comparison. We further extract seven tasks from RULER32K into RULER32K-HARD to isolate cases where top-$k$ methods are known to struggle.

Table 2: AIME@2024 with deepseek-ai/DeepSeek-R1-Distill-Llama-8B with vAttention ($\epsilon = 0.05$, $\delta = 0.05$, $f_t = 0.025$, $f_b = 0.025$, sink and local tokens are set to absolute 128). The generations are capped at 32K tokens. More details on the evolution of density and errors along sequence length are provided in Appendix A.5. Average density at 16K length is around 10-15% for HashAttention.

| Type | 1 | 2 | 3 | 4 | avg |
|---|---|---|---|---|---|
| dense | 43.30 | 36.67 | 33.33 | 33.33 | 36.66 |
| vAttention(oracle-top-$k$) | 43.33 | 40.00 | 26.66 | 36.67 | 36.67 |
| vAttention(HashAttention) | 30.00 | 36.66 | 46.66 | 26.66 | 35.00 |

RULER32K-HARD consists of `qa_1`, `qa_2`, `vt`, `fwe`, `niah_multikey_2`, `niah_multikey_3`, and `niah_multivalue`, selected based on the HashAttention paper, where these datasets were shown to be challenging. Detailed results are provided in the Appendix, with partial results included here. We also use AIME2024 to evaluate vAttention on long generation and reasoning tasks. For models, we consider three LLMs: Llama-3.1-8B-Instruct, Mistral-7B-v0.3, and DeepSeek-R1-Distill-Llama-8B, evaluated across different subsets of benchmarks.

**Baselines.** In our study, we choose *oracle-top-$k$* as a baseline, which serves as the theoretical gold standard for all approximate top-$k$ methods, and *oracle-top-$p$*, the strongest oracle-top-based baseline, as it can provide a dynamic oracle-top-$k$ based on the attention score distribution. As a representative of approximate top-$k$ methods, we select HashAttention, which outperforms Quest Tang et al. (2024), Double Sparsity(Yang et al., 2024), InfLLM Xiao et al. (2024), and others. For completeness, a comparative table is provided in Appendix A.4. We report results for vAttention in combination with both oracle-top-$k$ and HashAttention. When aiming to achieve a particular sparsity in an experiment, we search for the best parameters within a defined search space that yield the lowest local attention errors while meeting the target sparsity. Details of the search space are provided in Table 3. Following (Desai et al., 2025; Jegou et al., 2024; Hooper et al., 2024), we use full attention for context processing and sparse attention for question and generation. Under this setup, MagicPig does not perform well. We provide additional details in Appendix C.

**Superior quality of vAttention at different sparsities**   The Pareto results comparing the quality of sparse attention across different densities (i.e., the number of tokens used) are presented in Figure 4. We also compare average improvements across models in Table 1. We observe:

- vAttention significantly improves the quality and approximation error versus density tradeoff when combined with a top-$k$ method (both HashAttention and oracle top-$k$). vAttention combined with oracle top-$k$ yields the best results, indicating that more accurate top-$k$ methods are essential for the overall quality of sparse attention, even with vAttention.
- Oracle top-$p$, representing the best achievable top-$k$ approximation, does not always reach full model quality or provide the best approximation error at reasonable sparsities. For example, on the RULER 32K benchmarks, vAttention combined with oracle top-$k$ even outperforms oracle top-$p$.
- vAttention combined with oracle top-$k$ achieves full model accuracy at reasonable sparsity levels across all benchmark datasets considered.
- vAttention increases the average quality of 10%-sparse HashAttention on RULER-HARD by 4.6 percentage points for Llama3.1-8B and by 4.3 points for Deepseek-R1-Distill-Llama-8B.

**Long Generation with vAttention in the wild:** The AIME2024 results are presented in Table 2. We deploy vAttention with natural configuration parameters, without any parameter tuning, as would be done in real-world settings. Token generation is capped at 32K tokens. We find that both vAttention (oracle-top-$k$) and vAttention (HashAttention) match the full model quality (Avg@4), demonstrating the effective long-sequence generation capacity of vAttention.

**Efficiency with vAttention:** We compare the speedups of Llama-3-8B and Llama-2-7B with vAttention when the KV cache is hosted on the CPU, as shown in Figure 5. For this comparison, we use a naive PyTorch implementation of vAttention index computation together with our optimized sparse attention backend. With a more careful implementation, the performance of vAttention can be further improved for CPU-hosted KV caches and will yield gains for GPU-hosted KV caches. However, developing optimized CUDA kernels for vAttention is beyond the scope of this paper.

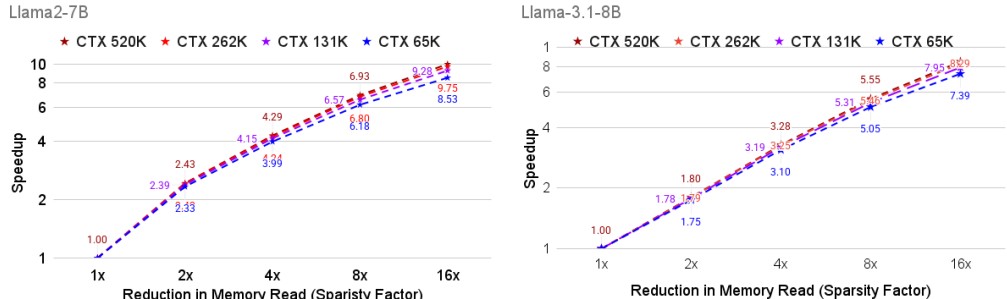

Figure 5: For Llama models with the KV cache hosted on the CPU, we observe a near-linear speedup, as inference is memory-bound and latency primarily depends on the amount of KV cache read. This experiment is conducted using naive PyTorch code for index computation, and the results can be further improved with a dedicated CUDA implementation.

## 6 CONCLUSION

The mainstream approach to sparse attention has been to approximate top-k token selection. However, as we show in this paper, even oracle versions of top-k and top-p are either insufficient for accurate attention approximation or require unnecessarily large numbers of tokens. More importantly, none of the existing methods—primarily designed to approximate these oracle versions—offer guarantees or user control over approximation errors. vAttention is the first verified sparse attention method that not only provides fine-grained user control over approximation, but also achieves superior quality–efficiency trade-offs compared to top-k approaches. It makes a compelling case for reliable deployment and for realizing the significant quality–efficiency benefits sparse attention can offer. We believe vAttention will enable the practical adoption of sparse attention in the wild.

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

# A    ADDITIONAL EXPERIMENTS RESULTS AND DETAILS

## A.1    PARETO PLOTS FOR LLAMA-3.1-8B-INSTRUCT

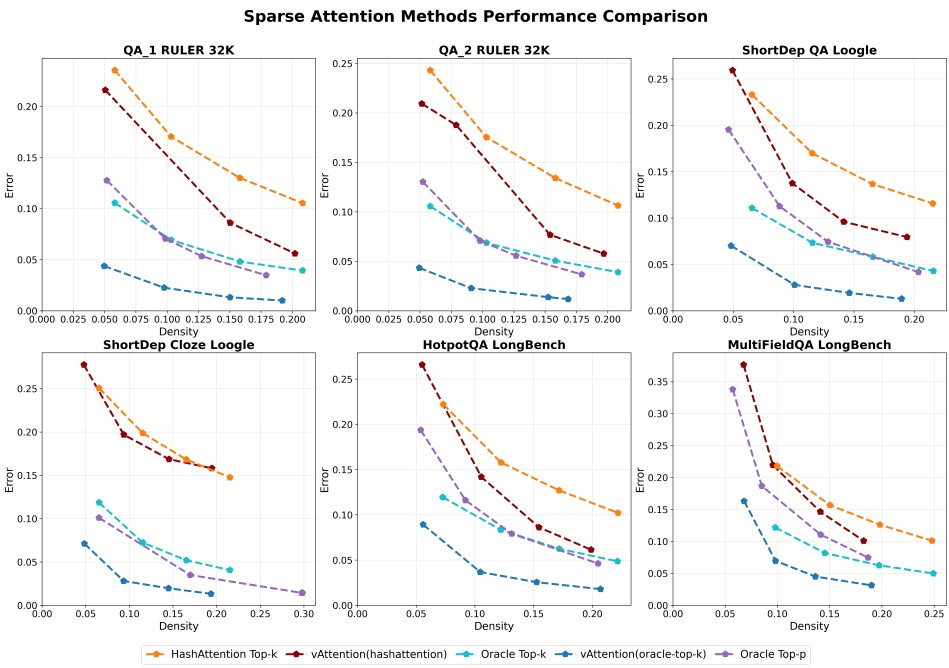

Figure 6: Attention approximation errors vs. Density for different approaches with and without vAttention.

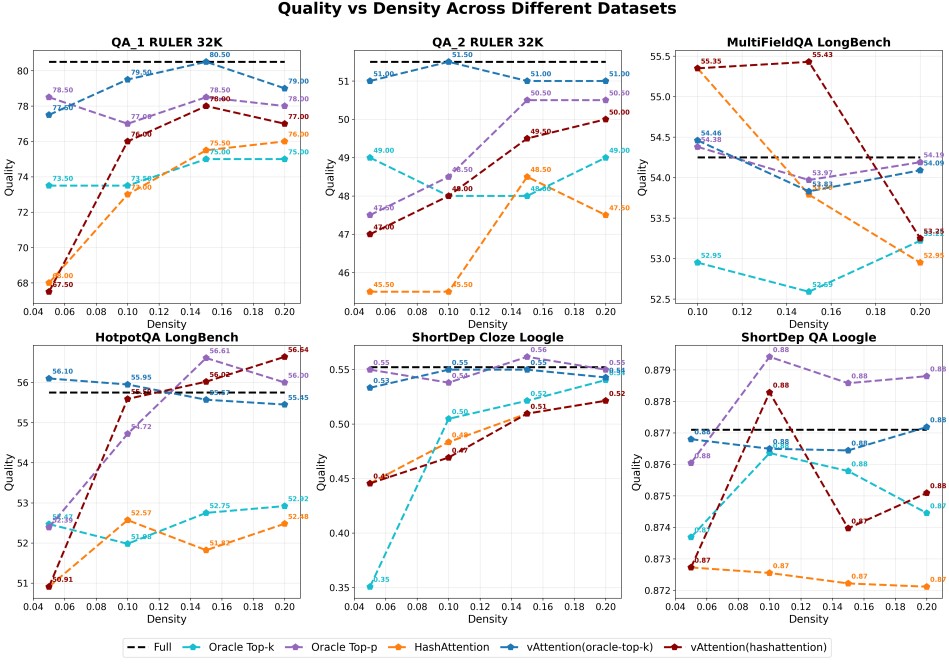

Figure 7: Quality vs. Density for different approaches with and without vAttention.

## A.2 ACHIEVING REQUIRED SPARSITY.

To achieve the required sparsity, we search for the configuration parameters on a few examples from the dataset and select the configuration with the minimum local attention approximation error while maintaining the target sparsity. The search space used is mentioned table 3,

Table 3: For local and sink tokens we use fixed 128 tokens throughout experiments for all methods.

|  | Target Sparsity | Parameter Grid |
|---|---|---|
| MagicPig |  | K = [4, 8, 16, 32] L = [16, 32, 64, 128] |
| oracle-top-$p$ |  | p=[0.1, 0.2, 0.3, 0.4, 0.5, 0.6, 0.7, 0.8, 0.85, 0.9, 0.95, 0.98, 0.99] |
| vAttention | 5 | $f_b$ : [0.01, 0.02, 0.03] 
 $f_t$: [0.05, 0.1, 0.15] 
 $\epsilon$: [0.05, 0.1, 0.2, 0.3] 
 $\delta$: [0.05, 0.1, 0.2, 0.3] |
| vAttention | 10 | $f_b$ : [0.025, 0.05, 0.075] 
 $f_t$: [0.05, 0.075, 0.1] 
 $\epsilon$:[0.025, 0.05, 0.1, 0.2] 
 $\delta$: [0.025, 0.05, 0.1, 0.2] |
| vAttention | 15 | $f_b$ : [0.025, 0.05, 0.075, 0.1] 
 $f_t$: [0.025, 0.05, 0.075] 
 $\epsilon$:[0.01, 0.025, 0.05, 0.1] 
 $\delta$: [0.01, 0.025, 0.05, 0.1] |
| vAttention | 20 | $f_b$ : [0.05, 0.1, 0.15] 
 $f_t$: [0.01, 0.02, 0.03] 
 $\epsilon$: [0.01, 0.025, 0.05, 0.1] 
 $\delta$: [0.01, 0.025, 0.05, 0.1] |

## A.3 DETAILED RESULTS AT 10% SPARSITY

Table 4: RULER @ 32K (Meta-llama/Llama-3.1-8B-Instruct) at 10% sparsity full benchmark results. As mentioned in the paper, half of the datasets are quite easy and solvable by HashAttention. However, the harder datasets is where vAttention bridges the gap between full attention and HashAttention / oracle-top-$k$ significantly.

| | niah_single_1 | niah_single_2 | niah_single_3 | niah_multikey_1 | niah_multiquery | niah_multivalue | cwe | vt | qa_1 | qa_2 | fwe | niah_multikey_2 | niah_multikey_3 | niah_multivalue |
|---|---|---|---|---|---|---|---|---|---|---|---|---|---|---|
| full attention | 100 | 100 | 100 | 100 | 97 | 98.5 | 1.6 | 97.4 | 80.5 | 51.5 | 93.17 | 99.5 | 100 | 99.12 |
| vAttention(oracle-top-$k$) | 100 | 100 | 100 | 100 | 97 | 98 | 1.2 | 97.5 | 79.5 | 51.5 | 93.17 | 99.5 | 100 | 99.12 |
| oracle-top-$k$ | 100 | 100 | 100 | 100 | 98.5 | 97.5 | 1.2 | 97.6 | 73.5 | 48 | 93.17 | 99.5 | 99.5 | 99 |
| vAttention(HashAttention) | 100 | 100 | 100 | 100 | 98 | 94 | 0 | 96.2 | 76 | 48 | 93.83 | 98.5 | 95 | 98.38 |
| HashAttention | 100 | 100 | 100 | 100 | 99 | 98 | 3.6 | 89 | 73 | 45.5 | 91.33 | 88.5 | 87.5 | 98.75 |

Table 5: RULER @ 32K (Meta-llama/Llama-3.1-8B-Instruct) average score

|  | Easy Average | Hard Average | Full Average |
|---|---|---|---|
| full attention | 85.30 | 88.74 | 87.02 |
| vAttention(oracle-top-$k$) | 85.17 | 88.61 | 86.89 |
| oracle-top-$k$ | 85.31 | 87.18 | 86.25 |
| vAttention(HashAttention) | 84.57 | 86.56 | 85.57 |
| HashAttention | 85.80 | 81.94 | 83.87 |

Table 6: [Context length capped at 32K and new tokens capped to 100 and 10% sparsity] Some datasets from Longbench benchmark with Llama-3.1-8B-Instruct

| dataset | multifieldqa_en | hotpotqa | narrativeqa | qasper | musique | qmsum | 2wiki | Avg |
|---|---|---|---|---|---|---|---|---|
| full attention | 54.25 | 55.75 | 29.25 | 46.87 | 31.07 | 24.98 | 46.51 | 41.24 |
| vAttention(oracle-top-$k$) | 54.46 | 55.95 | 29.17 | 47.79 | 30.57 | 25.10 | 46.27 | 41.33 |
| oracle-top-$k$ | 52.95 | 51.98 | 30.01 | 44.48 | 23.29 | 25.45 | 47.28 | 39.35 |
| vAttention(HashAttention) | 53.21 | 55.59 | 31.34 | 43.05 | 27.58 | 25.15 | 44.33 | 40.04 |
| HashAttention | 53.79 | 52.57 | 27.15 | 43.96 | 23.13 | 24.38 | 45.50 | 38.64 |

Table 7: RULER @ 32K (deepseek-ai/DeepSeek-R1-Distill-Llama-8B) at 10% sparsity full benchmark results. We only evaluate hard datasets for this setting

| dataset | vt | qa_1 | qa_2 | fwe | niah_multikey_2 | niah_multikey_3 | niah_multivalue | Avg |
|---|---|---|---|---|---|---|---|---|
| full attention | 21.20 | 46.00 | 58.00 | 90.67 | 82.00 | 68.00 | 92.00 | 65.41 |
| vAttention(oracle-top-$k$) | 18.40 | 44.00 | 60.00 | 90.67 | 82.00 | 70.00 | 91.00 | 65.15 |
| oracle-top-$k$ | 35.60 | 42.00 | 58.00 | 88.00 | 74.00 | 66.00 | 90.50 | 64.87 |
| vAttention(HashAttention) | 30.40 | 46.00 | 56.00 | 90.00 | 78.00 | 64.00 | 91.00 | 65.06 |
| HashAttention | 40.40 | 36.00 | 56.00 | 84.00 | 70.00 | 50.00 | 88.50 | 60.70 |

Table 8: RULER @ 32K (mistralai/Mistral-7B-Instruct-v0.3) at 10% sparsity full benchmark results. We only evaluate hard datasets for this setting

| dataset | vt | qa_1 | qa_2 | fwe | niah_multikey_2 | niah_multikey_3 | niah_multivalue | Avg |
|---|---|---|---|---|---|---|---|---|
| full attention | 88.00 | 50.00 | 48.00 | 91.33 | 60.00 | 20.00 | 91.00 | 64.05 |
| vAttention(oracle-top-$k$) | 88.00 | 52.00 | 44.00 | 91.33 | 60 | 24 | 89.5 | 64.12 |
| oracle-top-$k$ | 84.40 | 56.00 | 40.00 | 90.67 | 54.00 | 34.00 | 91.50 | 64.37 |
| vAttention(HashAttention) | 85.60 | 46.00 | 44.00 | 90.67 | 36 | 4 | 92 | 56.90 |
| HashAttention | 72.80 | 46.00 | 38.00 | 93.33 | 34.00 | 4 | 94.50 | 54.66 |

## A.4 COMPARATIVE RESULTS OF HASHATTENTION VS. OTHERS

The results are presented in Table 9

Table 9: Comparison of approximate top-k baselines with HashAttention on datasets from LongBench for LLama-3.1-8B-Instruct. All baselines are used at 32 bits per token per head of auxilliary memory. The table is adapted from (Desai et al., 2025)

| Model | Category →
Aux:bits/token | Tokens | MQA
HPQA | SQA
MFQA | Summ
QmSm | FS-Learn
TQA | Synthetic
PassR | Code
RepoB | Average |
|---|---|---|---|---|---|---|---|---|---|
| Full Model | NA | NA | 54.83 | 55.17 | 24.91 | 91.31 | 100.00 | 55.07 | 63.55 |
| Oracle(top) | NA | 512 | 52.10 | 53.45 | 25.14 | 91.39 | 100.00 | 58.49 | 63.43 |
| H2O | NA | 512 | 36.62 | 26.61 | 17.85 | 80.75 | 43.43 | 55.55 | 43.47 |
| StreamLLM | NA | 512 | 33.32 | 27.98 | 17.93 | 51.95 | 11.43 | 57.07 | 33.28 |
| InfLLM | 256(pg=16,bit=16) | 512 | 48.27 | 53.09 | 22.90 | 88.88 | 32.81 | 43.45 | 48.23 |
| DS | 32(ch=16,bit=2) | 512 | 50.39 | 50.57 | 23.41 | 90.32 | 98.86 | 57.72 | 61.88 |
| Quest | 32(pg=16,bit=2) | 512 | 53.13 | 51.31 | 23.01 | 90.15 | 98.29 | 58.29 | 62.36 |
| HashAttention | 32 | 512 | **54.08** | 53.35 | **25.08** | **92.41** | **100.00** | **59.98** | **64.15** |

## A.5 LONG GENERATION WITH VATTENTION

The error and density evolution with token generation for AIME for two examples in Figures 8, 9. vAttention adapts the sparsity in each layer for each head and for each specific query. The average density of attention at 32K tokens is around 12%. Even with natural parameter values for parameters of vAttention, it can achieve the required density, which leads to stable long generation.

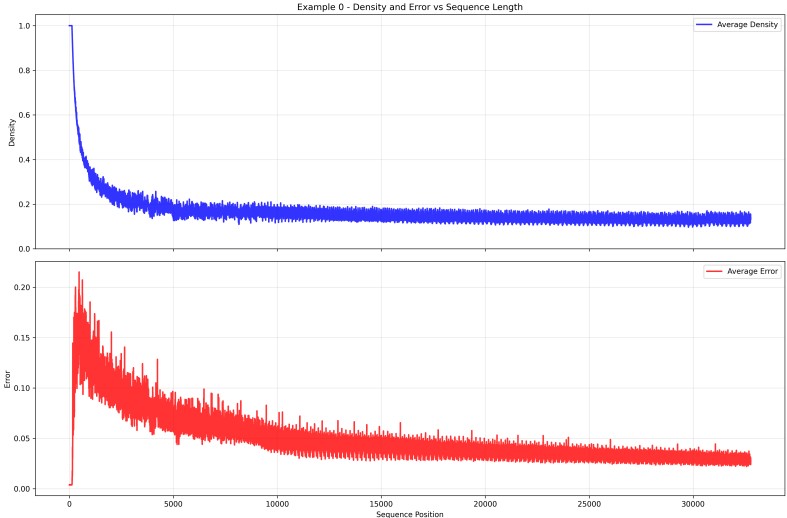

Figure 8: Example 0

## A.6 ABLATION $(\epsilon, \delta)$

# B DETAILED RELATED WORK

Existing work on sparse attention can be categorized into the following types, covering early explorations to recent efforts.

## B.1 STATIC SPARSE ATTENTION

Early work on sparse attention focused on fixed sparsity patterns to reduce the number of tokens considered during decoding. For instance, StreamingLLM (Xiao et al., 2023) employs fixed attention

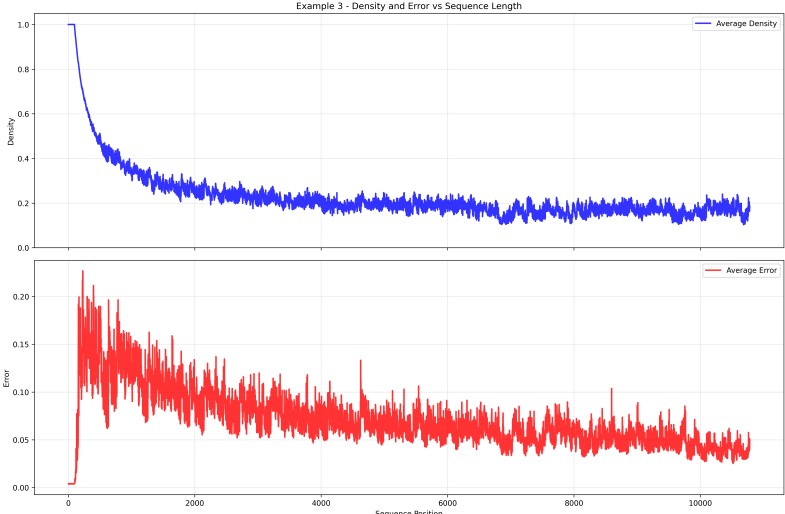

Figure 9: Example 3

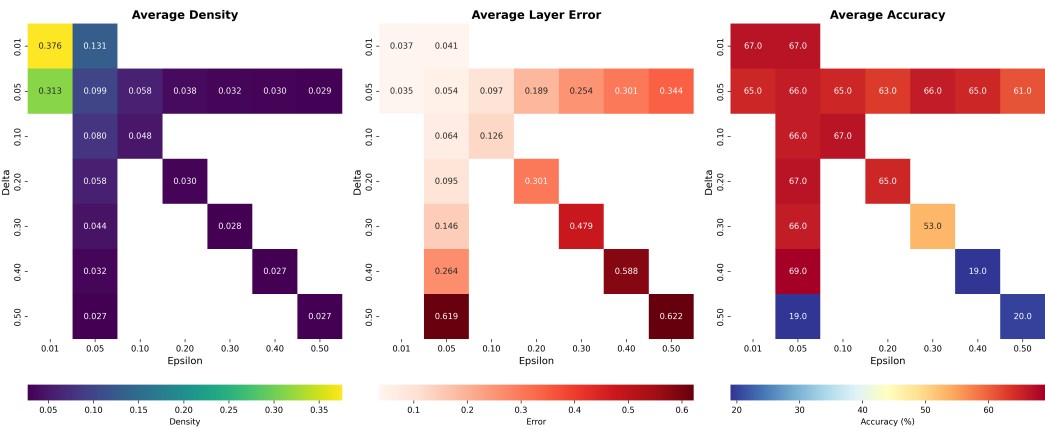

Figure 10: How the average density, average relative attention approximation error and overall quality on RULER-32K(QA1, QA2) tasks vary when using only denominator-guaranteed approximation

to both "attention sinks" (typically the first few tokens) and a sliding window over recent tokens. However, subsequent studies (Zhang et al., 2023; Xiao et al., 2024) have shown that static patterns often fail to generalize well, highlighting the importance of dynamic sparsity in attention mechanisms. While StreamingLLM itself does not fully support dynamic token selection, its key insight—that attention sinks and local windows are crucial—has strongly influenced later approaches. Most recent sparse attention methods now incorporate sink and local tokens as foundational components of their selection strategies.

## B.2 KV CACHE COMPRESSION METHODS

A parallel line of research focuses on KV cache compression, wherein tokens are selectively discarded based on heuristics aimed at reducing memory overhead. Representative approaches include ScissorHands (Liu et al., 2024b), H2O (Zhang et al., 2023), FastGen (Ge et al., 2023), and SnapKV (Li et al., 2024a). While these methods can be effective in lowering memory usage, they often lack generalizability across tasks due to their irreversible token pruning. This limitation is particularly pronounced in settings such as multi-turn dialogue/interaction, where the relevance of contextual tokens may vary significantly between turns, making fixed pruning strategies inadequate.

### B.3 APPROXIMATE TOP-$k$ BASED SPARSE ATTENTION

A line of sparse attention methods is grounded in the observation that only a small subset of tokens significantly contribute to attention computation—typically those associated with the highest attention scores under full attention. For a given query vector $q$, the most relevant tokens are those whose key vectors $k$ yield the highest inner products $q^\top k$. In theory, optimal sparsity could be achieved by selecting the top-$k$ tokens with the highest inner products. However, computing all pairwise inner products requires $O(nd)$ operations, which diminishes the potential computational savings. Consequently, most methods in this class rely on various approximations to efficiently estimate the top-$k$ tokens.

Recent methods in this direction introduce specific strategies to reduce the computational costs of inner products. For instance, Double Sparsity (Yang et al., 2024) approximates inner products using a reduced set of channels, while Loki (Singhania et al., 2024) leverages low-rank projections to operate in a compressed, lower-dimensional space. Other techniques, such as InfLLM (Xiao et al., 2024) and Quest (Tang et al., 2024), employ page-level summary vectors to identify potentially important tokens at the block level, thereby limiting the number of inner products evaluated. Despite their computational benefits, these methods often rely on heuristic approximations, which can lead to poor recall in identifying the true top-$k$ tokens critical for accurate attention computation.

Given that the task of identifying top-$k$ tokens effectively reduces to the Maximum Inner Product Search (MIPS) problem (Desai et al., 2025), a number of recent methods have adopted techniques from approximate nearest neighbor (ANN) search. For example, PQCache (Zhang et al., 2025) leverages product quantization to accelerate MIPS, while SqueezeAttention (Hooper et al., 2024) employs hierarchical clustering to improve the efficiency of top-$k$ retrieval. Retrieval Attention (Li et al., 2024b) adopts graph-based ANN search, and HashAttention (Desai et al., 2025) encodes queries and keys as bit signatures, enabling efficient similarity computation in Hamming space.

Although these approaches improve scalability by narrowing the search to the most promising tokens, their reliance on approximating the oracle top-$k$ tokens introduces a fundamental limitation. As shown in MagicPig (Chen et al., 2024), and further analyzed in this work, even access to the exact top-$k$ tokens under full attention does not always suffice to faithfully approximate the original attention output—highlighting the need to go beyond top-$k$ selection in designing effective sparse attention mechanisms.

### B.4 APPROXIMATE TOP-$p$ BASED SPARSE ATTENTION

A key limitation observed in top-$k$-based sparse attention methods is that a fixed sparsity level fails to generalize across different attention modules within a model. To address this, recent approaches have shifted towards achieving top-$p$ coverage, where the goal is to select a variable number of tokens whose cumulative attention scores under full attention exceed a threshold $p$. This adaptive strategy better aligns with the varying importance distributions across layers and heads. Additionally, it provides control over the amount of error an attention module can make.

However, identifying the exact set of tokens that satisfy the top-$p$ criterion—i.e., those whose cumulative attention scores exceed a predefined threshold $p$—is computationally more demanding than top-$k$ selection, as it requires sorting or aggregating over all token scores. To mitigate this cost, recent methods approximate the coverage estimation to efficiently select token indices that collectively capture the desired attention mass. One such approach is Tactic (Zhu et al., 2025), which approximates top-$p$ attention by modeling the decay of attention scores using a power-law distribution, allowing for efficient estimation of how many top-scoring tokens are needed to meet the coverage threshold.

As we will show in this paper, while top-$p$ attention offers some degree of error control—subject to the quality of its approximation—it is not the most efficient approach for achieving a given error bound. More principled mechanisms can attain comparable or lower error using fewer tokens. In this work, we introduce one such method: vAttention, which enables improved error control through adaptive and token-efficient selection.

### B.5 MagicPig: LSH sampling based Sparse Attention

To the best of our knowledge, MagicPig was the first work to highlight the issues associated with top-$k$-based sparse attention.

The method leverages Locality Sensitive Hashing (LSH) (Gionis et al., 1999) to select which tokens participate in attention computation. While LSH is generally considered suboptimal for approximate nearest neighbor (ANN) search due to its data-agnostic projections, its use here offers a principled and novel mechanism for approximating attention. LSH-based retrieval can be viewed as a sampler (Luo & Shrivastava, 2018). Thus, the tokens retrieved from LSH have probabilities associated with them, under which they were sampled in the randomized construction of the LSH table. We can estimate the numerator and denominator of attention using the importance sampling formulation. Early exploration for vAttentionwas inspired by MagicPig, and we will elaborate more on the attention computation in subsequent sections.

While LSH gives a principled way to compute attention, the issue associated with using LSH remains. Firstly, given the orthogonal distribution of keys and queries (Chen et al., 2024), LSH fails to distinguish between the different keys to the level at which original softmax demands – often leading to a highly skewed distribution of buckets (some buckets are very heavy while other buckets are empty). Centering is considered to be a practical solution to this. However, it is easy to prove that under centering, the original ordering among tokens for an arbitrary query is not preserved, making the operation ad hoc. Even if centering is valid, the number of hashes required to achieve sufficient recall is significantly higher than that of related methods, such as HashAttention(Desai et al., 2025), necessitating the involvement of CPU RAM.

Finally, none of the existing methods across categories offer concrete guarantees on the quality of approximation—even at the level of a single attention head. In contrast, vAttention addresses this gap by providing a principled solution to the problem of uncontrolled approximation in sparse attention. Our method enables explicit control over the approximation quality for each individual attention head, offering both reliability and flexibility.

## C    Evaluation Setup and Comparison on MagicPig

Since our work is primarily concerned with the efficiency of sparse attention during decoding, it is common practice to preprocess long contexts using full attention. Under this paradigm, two evaluation setups for sparse attention are typically employed:

1. **[Setup A] Full-prompt preprocessing with dense attention followed by sparse decoding**
   The entire prompt is first processed with full attention, and sparse attention is applied only during the decoding phase. In this setup, the first token generated already benefits from full attention.

2. **[Setup B] Split-prompt processing (context vs. question)**: The prompt is divided into two parts:
   (a) Context is processed with full attention
   (b) Question + subsequent generations are processed with sparse attention

Some earlier works, such as MagicPig, adopt the first setup. In contrast, more recent approaches—including HashAttention and SqueezeAttention—follow the second. A methodology similar to the second is also used in NVIDIA's KVPress – a framework to compare KV cache compression methods, where the KV cache is compressed after the context is processed but before the question is introduced. We argue that the second setup is the more meaningful choice.

The reasoning is as follows. Sparse attention for long-context evaluation is usually tested on datasets with relatively short generations (e.g., RULER, LongBench, etc). Suppose the entire context is first processed by full attention (setup 1). In that case, all the necessary information to answer the question has already been extracted by the time the first token is predicted. Applying sparse attention only after this point, especially with a fixed local attention window, does not truly test its ability to retrieve and utilize information from the long context. This hypothesis is validated by observations where, under setup A, MagicPig appears to perform well, but their performance collapses under setup B. (see Table 10)

Therefore, to genuinely assess the effectiveness of sparse attention in long-context settings, it is essential to adopt setup B

Table 10: Faithful reproduction of MagicPig results on RULER and differences from our MagicPig implemenation and evaluation setup. **Evaluation-A**: preprocess full context+question via full attention followed by sparse attention only for genreation. **Evaluation-B**: preprocess only context with full attention and question along with generations are processed by sparse attention. **MagicPig-A**: (logic in Authors Code) which does not use simpleLSH transform for inner product search (only uses angular LSH) and uses dense layers for 0,16. **MagicPig-B**(logic of our base code) uses simpleLSH transform for inner product search as per theory, does not use any dense layers. In our evaluation setup, which is a more reasonable evaluation, MagicPig does not perform well.

| | setup | MagicPig (K=8,L=75) | niah_single_1 | niah_single_2 | niah_single_3 | niah_multikey_2 | niah_multikey_3 | niah_multivalue |
|---|---|---|---|---|---|---|---|---|
| Authors code | A = B
+ questions processed via
dense attention | A =B (core paper description)
+ dense layers(0,16)
+ no simpleLSH transform | 100 | 100 | 100 | 98 | 98 | 98 |
| Our code | **B** | **B** | **100** | **96** | **76** | **46** | **12** | **81.5** |
| | B | B
+ dense layers(0,16) | 100 | 96 | 96 | 74 | 60 | 84.5 |
| | A | B
+ dense layers(0,16) | 100 | 98 | 98 | 94 | 90 | 88 |
| | A | A | 100 | 100 | 100 | 98 | 98 | 95.5 |

Furthermore, some methods deliberately mix dense and sparse attention across layers. We take a different stance: sparse attention itself should be sufficiently adaptive to each layer's requirements, increasing its effective density when necessary rather than relying on dense layers as a fallback.

## D  THEORY

### D.1  DERIVATION VIA CENTRAL LIMIT THEOREM (CLT)

**Lemma D.1 (Estimating vector sum).** *Let* $\mathbf{s} = \sum_{i=1}^{n_s} \mathbf{r}_i, \mathbf{s} \in R^d$ *be a sum of* $n_s$ *vector quantities* $\mathbf{r}_i \in R^d \, \forall i$ *which have to be estimated using a sample* $\mathcal{I}_b$ *of size b. Let* $\Sigma$ *be the covariance matrix for the population* $\{\mathbf{r}_i\}_{i=1}^{n_s}$. *Let* $\hat{\mathbf{s}}_b = \frac{n_s}{b} \left( \sum_{i \in \mathcal{I}_b} \mathbf{r}_i \right)$ *be the estimate. Let* $\Phi$ *be the CDF for the normal distribution. Then for a large enough b if,*

$$b \geq \left( \Phi^{-1} \left( 1 - \frac{\delta}{2} \right) \frac{n_s \sqrt{\mathbf{Tr}(\Sigma)}}{\tau} \right)^2 \quad then \quad \mathbf{Pr}(||\hat{\mathbf{s}} - \mathbf{s}||_2 > \tau) \leq \delta \qquad (11)$$

*for any arbitrary* $\tau \in R$ *and* $\delta \in (0,1)$.

Using the Multivariate Central Limit Theorem,

$$\sqrt{b} \left( \frac{1}{b} \sum_{i \in \mathcal{I}_b} \mathbf{r_i} - \frac{s}{n} \right) \xrightarrow{d} \mathcal{N}(\mathbf{0}, \Sigma).$$

$$\frac{\sqrt{b}}{n} \left( \frac{n}{b} \sum_{i \in \mathcal{I}_b} \mathbf{r_i} - s \right) \xrightarrow{d} \mathcal{N}(\mathbf{0}, \Sigma).$$

$$\frac{\sqrt{b}}{n} \left(\hat{\mathbf{s}} - \mathbf{s}\right) \xrightarrow{d} \mathcal{N}(\mathbf{0}, \mathbf{\Sigma}).$$

$$\frac{\sqrt{b}}{n} \mathbf{u}^\top \left(\hat{\mathbf{s}} - \mathbf{s}\right) \xrightarrow{d} \mathcal{N}(\mathbf{0}, \mathbf{u}^\top \mathbf{\Sigma} \mathbf{u}).$$

$$\mathbf{u}^\top \left(\hat{\mathbf{s}} - \mathbf{s}\right) \xrightarrow{d} \frac{n}{\sqrt{b}} \mathcal{N}(\mathbf{0}, \mathbf{u}^\top \mathbf{\Sigma} \mathbf{u}).$$

To achieve a error within $\tau$ for $\delta$, we must have

$$\frac{n}{\sqrt{b}} \sqrt{(\mathbf{u}^\top \mathbf{\Sigma} \mathbf{u})} \Phi^{-1} \left(1 - \frac{\delta}{2}\right) < \tau$$

This can be achieved if

$$\frac{n}{\sqrt{b}} (\sqrt{\mathbf{Tr}(\mathbf{\Sigma})}) \Phi^{-1} \left(1 - \frac{\delta}{2}\right) < \tau$$

Solving for $b$

$$b > \left(\frac{n}{\tau} (\sqrt{\mathbf{Tr}(\mathbf{\Sigma})}) \Phi^{-1} \left(1 - \frac{\delta}{2}\right)\right)^2$$

## D.2 Corollaries for Numerator and Denominator

**Corollary D.2** (($\epsilon, \delta$) **approximation of N**). *Let $\Sigma$ be the covariance matrix for the population* $\{\exp \langle K[i], q \rangle V[i]\}_{i \in \bar{\mathcal{I}}_f}$. *Let $\hat{N} = N_f + \frac{n_s}{b} \left(\sum_{i \in \mathcal{I}_{dyn}} \exp \langle K[i], q V[i] \rangle\right)$ be the estimate when using sample $\mathcal{I}_{dyn}$ of size $b$. Let $\Phi$ be the CDF for the normal distribution. Then for a large enough $b$ and for any arbitrary $\epsilon, \delta \in (0, 1)$, if*

$$b \geq \left(\Phi^{-1} \left(1 - \frac{\delta}{2}\right) \frac{n_s \sqrt{\mathbf{Tr}(\Sigma)}}{\epsilon ||N||_2}\right)^2 \quad then \quad \mathbf{Pr}(||\hat{N} - N||_2 > \epsilon ||N||_2) \leq \delta \qquad (12)$$

**Corollary D.3** (($\epsilon, \delta$) **approximation of D**). *Let $\sigma$ be the standard deviation for the population* $\{\exp K[i], q\}_{i \in \bar{\mathcal{I}}_f}$. *Let $\hat{D} = D_f + \frac{n_s}{b} \left(\sum_{i \in \mathcal{I}_{dyn}} \exp \langle K[i], q \rangle\right)$ be the estimate when using sample $\mathcal{I}_{dyn}$ of size $b$. Let $\Phi$ be the CDF for the normal distribution. Then for a large enough $b$, for any arbitrary $\epsilon, \delta \in (0, 1)$, if*

$$b \geq \left(\Phi^{-1} \left(1 - \frac{\delta}{2}\right) \frac{n_s \sigma}{\epsilon D}\right)^2 \quad then \quad \mathbf{Pr}(|\hat{D} - D| > \epsilon D) \leq \delta \qquad (13)$$

## D.3 Combination of approximations of numerator and denominator

**Lemma D.4.** *If $b_D$ and $b_N$ are chosen such that we have $(\epsilon_1, \delta_1)$ and $(\epsilon_2, \delta_2)$ approximation on numerator and denominator respectively and $\epsilon_2 < 0.5$, then using $b = \max(b_D, b_N)$ ensures that*

$$\mathbf{Pr} \left(\left\|\frac{N}{D} - \frac{\hat{N}}{\hat{D}}\right\|_2 > 2(\epsilon_1 + \epsilon_2)\right) \left\|\frac{N}{D}\right\|_2 < (\delta_1 + \delta_2) \qquad (14)$$

If we have a $(\epsilon_1, \delta_1)$ approxiamtion for numerator and $(\epsilon_2, \delta_2)$ for denominator. Consider the following expression

$$||\frac{\hat{N}}{\hat{D}} - \frac{N}{D}||_2 = ||\frac{D\hat{N} - \hat{D}N}{\hat{D}D}|| \qquad (15)$$

With probability $(1 - \delta_1 - \delta_2)$

$$||\frac{D\hat{N} - DN \pm \epsilon_2 DN}{\hat{D}D}|| = ||\frac{\hat{N} - N \pm \epsilon_2 N}{\hat{D}}|| \tag{16}$$

$$\leq \frac{||\hat{N} - N||_2 + \epsilon_2||N||_2}{D(1 \pm \epsilon_1)} \tag{17}$$

$$\leq \frac{\epsilon_1||N||_2 + \epsilon_2||N||_2}{D(1 \pm \epsilon_2)} \tag{18}$$

$$= (\epsilon_1 + \epsilon_2)\frac{||N||_2}{D(1 - \epsilon_2)} \tag{19}$$

$$\leq (\epsilon_1 + \epsilon_2)(1 + 2\epsilon_2)\frac{||N||_2}{D} \quad \text{if } \epsilon_2 < 0.5 \tag{20}$$

$$\leq 2(\epsilon_1 + \epsilon_2)\frac{||N||_2}{D} \tag{21}$$

$$\tag{22}$$

## D.4    WHY REDUCING BIAS IN ESTIMATION IS MORE IMPORTANT THAN REDUCING VARIANCE.

The propagation of errors through the model can be modeled as a random walk of $\pm\epsilon_i$ steps for simplicity.

The argument follows from the standard analysis of mean-square error of random walk of $n$ steps. Let each step have a mean square error of size $\epsilon^2$.

**case 1: Entire MSE is attributed to bias**    Then the MSE at step $n$ is $n^2\epsilon^2$

**case 2: Entire MSE is attributed to variance**    Then the MSE at step $n$ is $n\epsilon^2$

Thus, the impact of bias on error propagation is much stronger than that of variance.

Generally, if the bias and standard deviation of error at each step are $\mu, \sigma$, then the MSE at step $n$ is

$$\text{MSE}(n) = n^2\mu^2 + n\sigma^2 \tag{23}$$

$$\text{MSE}(n) = (n(n - 1))\mu^2 + n\epsilon^2 \tag{24}$$

Thus the compounding effect of bias is much stronger than that of variance.

# E EMPIRICAL ANALYSIS OF TIGHTNESS FOR CLT AND HOEFFDING BOUNDS

In this section, we present an empirical analysis of the tightness of our theoretical bounds for denominator approximation. We compare the Central Limit Theorem (CLT) based approximation with Hoeffding's inequality, focusing on the configuration $\epsilon = 0.1, \delta = 0.2$ with 5% oracle top-$k$ selection.

## E.1 SUMMARY ANALYSIS ACROSS LAYERS

The results are presented in Figure 11 and Figure 12.

We evaluated the tightness of both bounds with the following setup:

- **Approximation parameters**: $\epsilon = 0.1$ and $\delta = 0.2$
- **Oracle top-$k$**: 5% of total tokens selected deterministically
- **Model**: Llama-3.1-8B-Instruct on RULER dataset with 16K context length

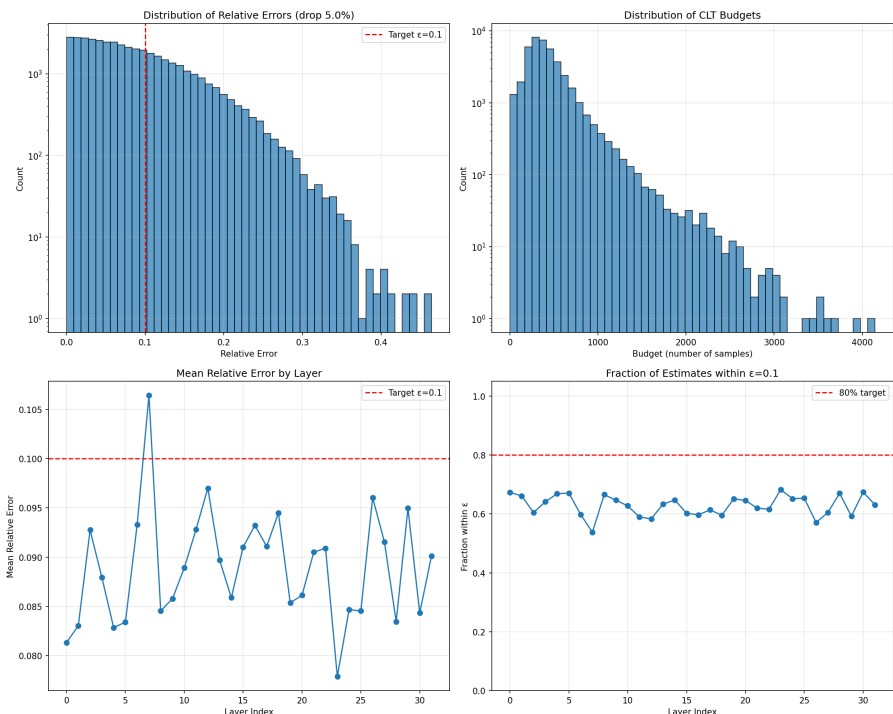

Figure 11: CLT-based approximation analysis with $\epsilon = 0.1, \delta = 0.2$ and 5% oracle top-$k$.

Comparing the two results (Figure 11 and Figure 12) , we find:

- **Conservative bounds**: Hoeffding equires 2.8× more samples (average 874) compared to CLT for the same guarantees.
- **Robust guarantees**: Hoeffing Achieves near-zero failure rate ($< 2\%$) but at significant computational cost.

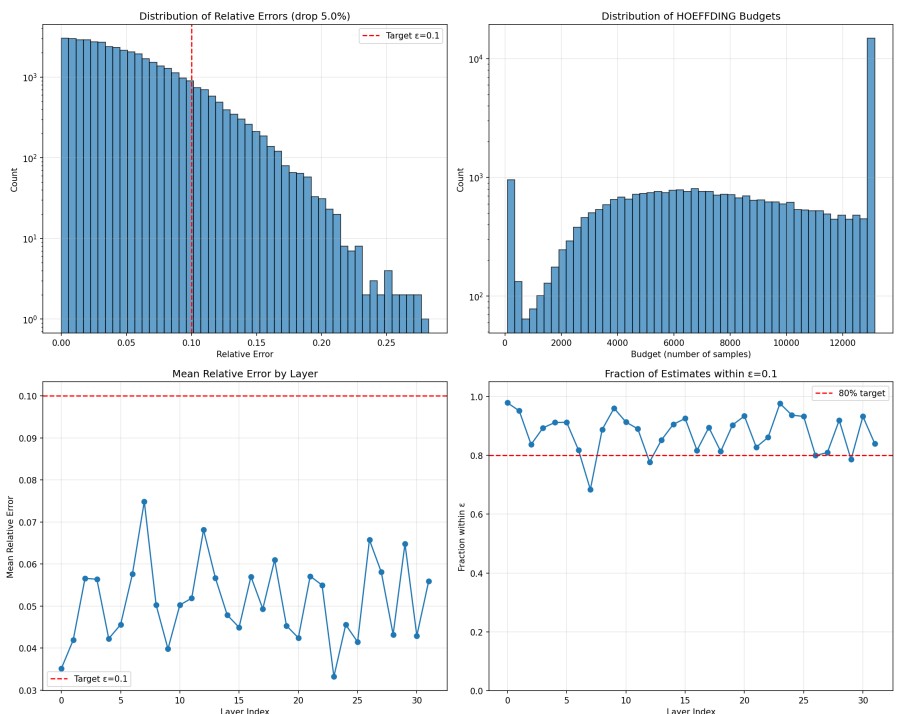

Figure 12: Hoeffding-based approximation analysis with $\epsilon = 0.1, \delta = 0.2$ and 5% oracle top-$k$, showing consistently higher sample requirements than CLT.

### E.2 LAYER-SPECIFIC ANALYSIS

For a given query, the distribution of attention scores $p_{l,h}$ varies significantly across different heads/layers. The adaptive design of vAttention allows for dynamic budget for a given tolerance level $(\epsilon, \delta)$. Across different layers (layer 1/16/32) of the Llama-3.1-8B-Instruct model, we measure the empirical budget for the CLT and Hoeffding-based budget estimates. In particular, for $(\epsilon = 0.1, \delta = 0.2)$, we investigate the distribution of relative errors and average budget/head in Figures 13, 14, 15.

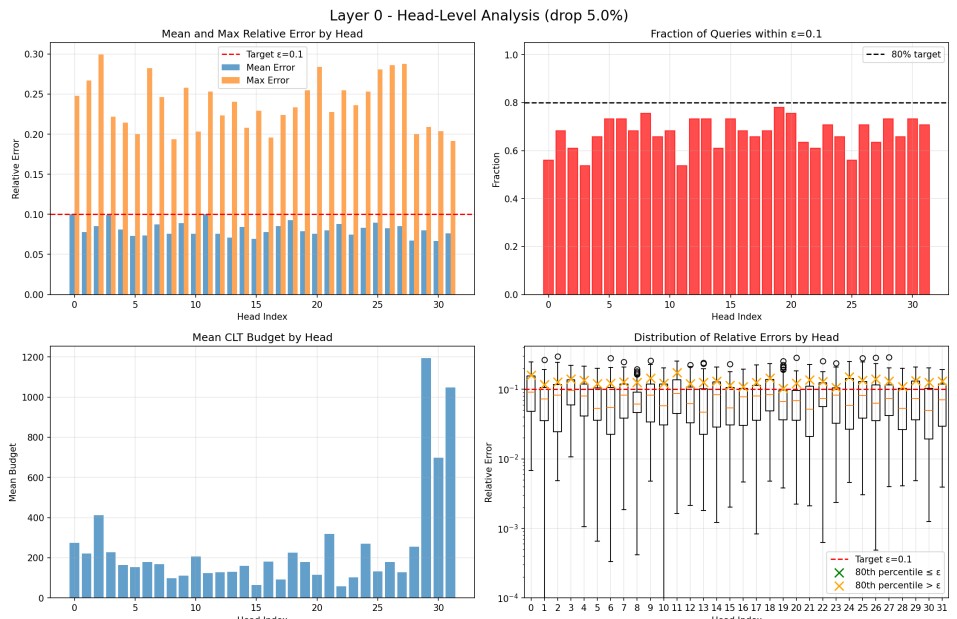

(a) **CLT budget** : **(top-left)** mean and maximum average relative error per head; the dashed line is the target error tolerance ($\epsilon$) **(top-right)** the fraction of queries ($\hat{\delta}$) that are within the specified error tolerance ($\epsilon$) **(bottom-left)** vAttention budget per head with CLT relaxation **(bottom-right)** distribution of relative errors ($\hat{\epsilon}$) across heads

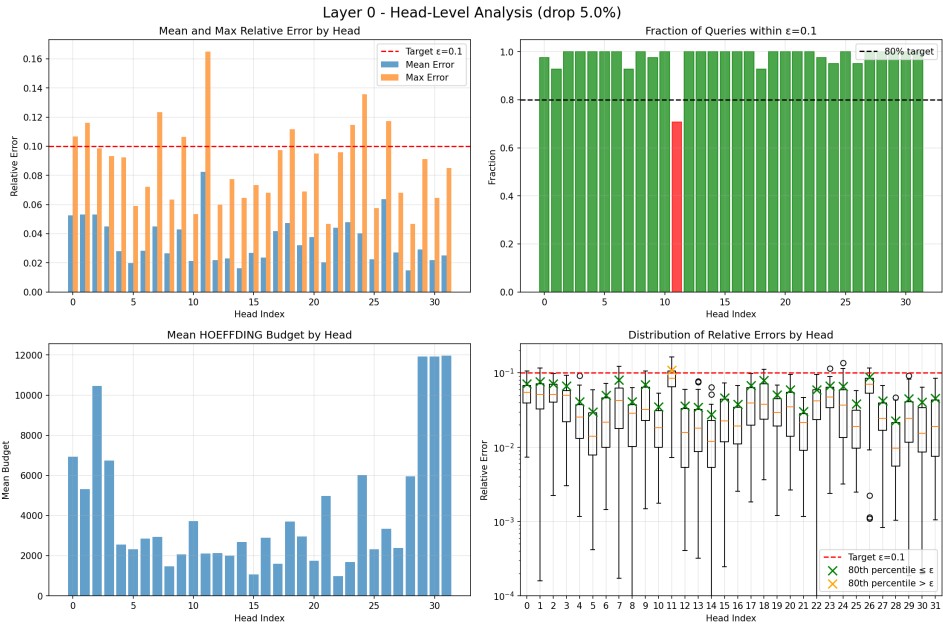

(b) **Hoeffding budget** : **(top-left)** mean and maximum average relative error per head; the dashed line is the target error tolerance ($\epsilon$) **(top-right)** the fraction of queries ($\hat{\delta}$) that are within the specified error tolerance ($\epsilon$) **(bottom-left)** vAttention budget per head with Hoeffding bound $\hat{b} << N$ **(bottom-right)** distribution of relative errors ($\hat{epsilon}$) across heads

Figure 13: **Layer 1 analysis**: For early layers, the Hoeffding budget is non-vacuous and is highly likely to meet the verification thresholds. Further, the budget with CLT relaxation leads to much smaller budgets while providing a decent likelihood with average relative error within the tolerance error ($\epsilon$)

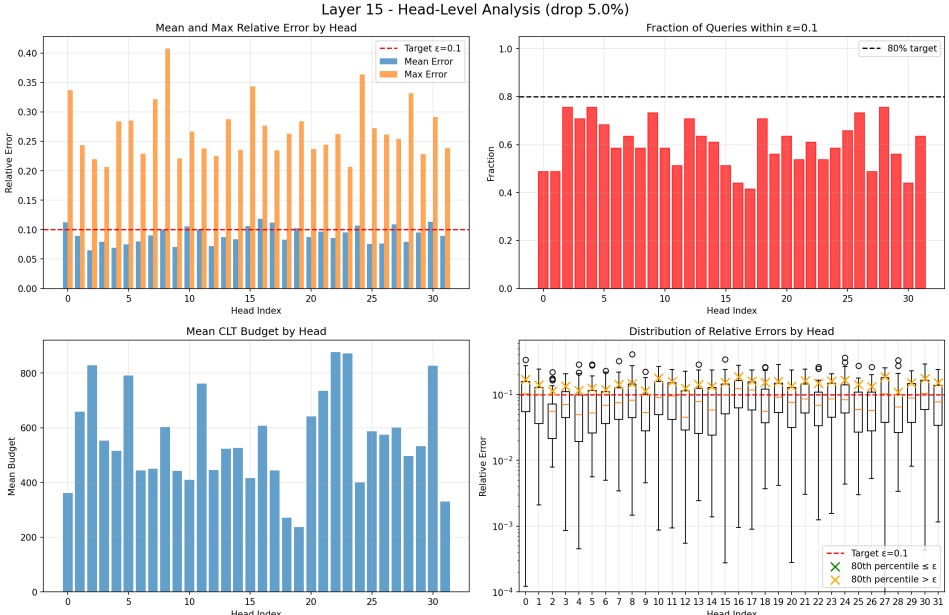

(a) **CLT budget** : **(top-left)** mean and maximum average relative error per head; the dashed line is the target error tolerance ($\epsilon$) **(top-right)** the fraction of queries ($\hat{\delta}$) that are within the specified error tolerance ($\epsilon$) **(bottom-left)** vAttention budget per head with CLT relaxation **(bottom-right)** distribution of relative errors ($\hat{\epsilon}$) across heads

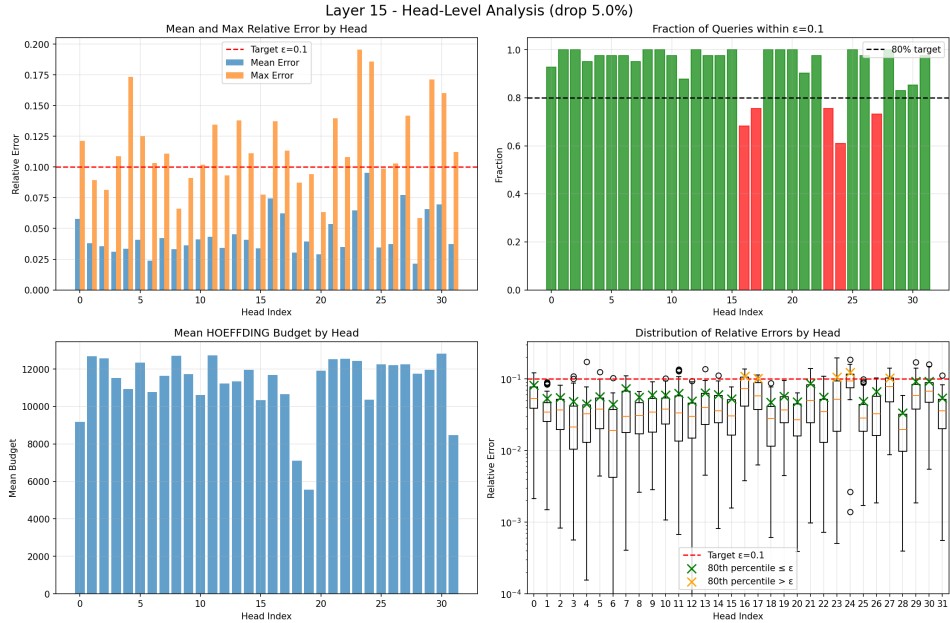

(b) **Hoeffding budget** : **(top-left)** mean and maximum average relative error per head; the dashed line is the target error tolerance ($\epsilon$) **(top-right)** the fraction of queries ($\hat{\delta}$) that are within the specified error tolerance ($\epsilon$) **(bottom-left)** vAttention budget per head with Hoeffding bound $\hat{b} << N$ **(bottom-right)** distribution of relative errors ($\hat{epsilon}$) across heads

Figure 14: **Layer 16 analysis**: For middle layers, the Hoeffding budget is more conservative and is requires high budget to meet the verification thresholds. Further, the budget with CLT relaxation leads to much smaller budgets while providing a decent likelihood with average relative error within the tolerance error ($\epsilon$), but have higher local errors

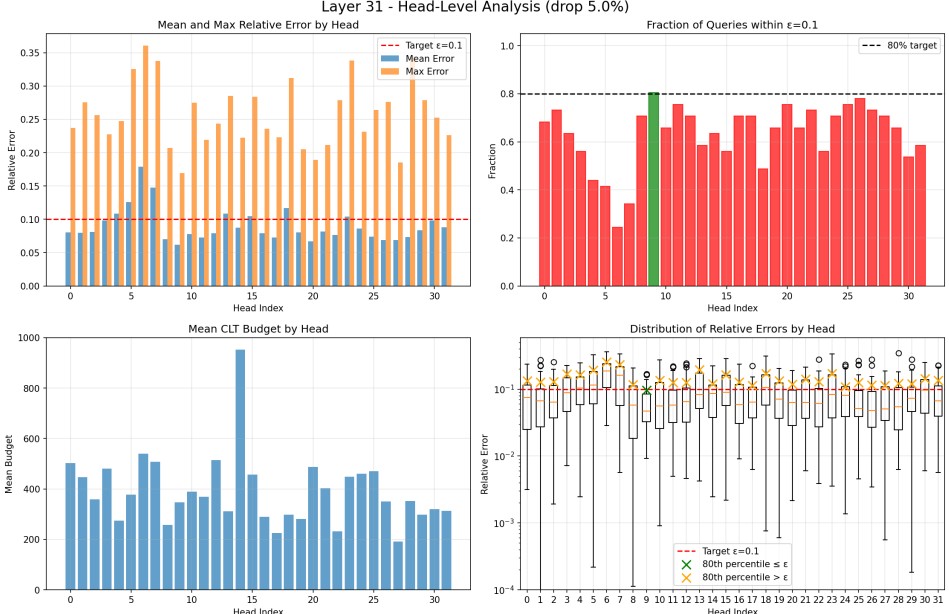

(a) **CLT budget** : **(top-left)** mean and maximum average relative error per head; the dashed line is the target error tolerance ($\epsilon$) **(top-right)** the fraction of queries ($\hat{\delta}$) that are within the specified error tolerance ($\epsilon$) **(bottom-left)** vAttention budget per head with CLT relaxation **(bottom-right)** distribution of relative errors ($\hat{\epsilon}$) across heads

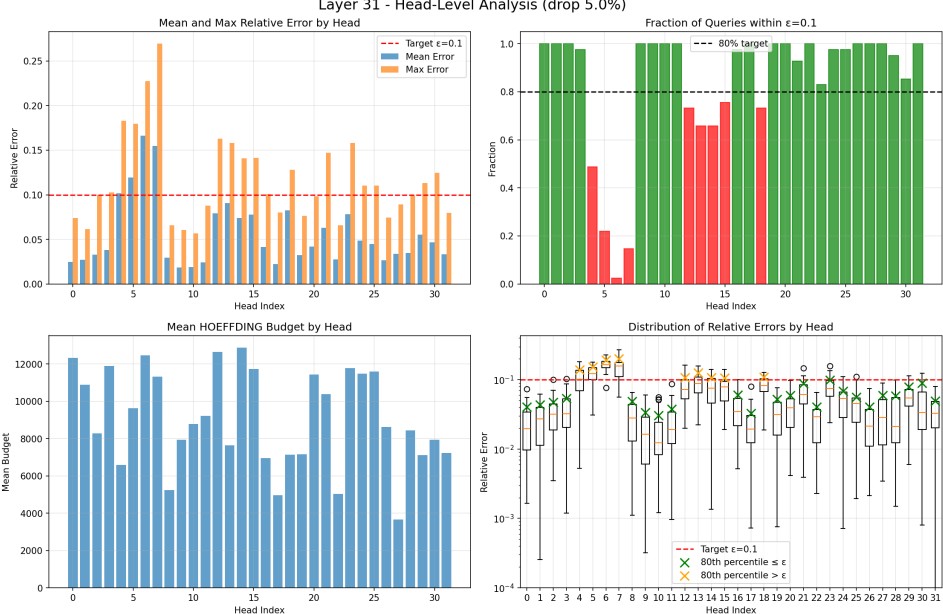

(b) **Hoeffding budget** : **(top-left)** mean and maximum average relative error per head; the dashed line is the target error tolerance ($\epsilon$) **(top-right)** the fraction of queries ($\hat{\delta}$) that are within the specified error tolerance ($\epsilon$) **(bottom-left)** vAttention budget per head with Hoeffding bound $\hat{b} << N$ **(bottom-right)** distribution of relative errors ($\hat{epsilon}$) across heads

Figure 15: **Layer 32 analysis**: For late layers, the Hoeffding budget is less likely to meet verification thresholds. Further, the budget with CLT relaxation leads to much smaller budgets while providing a decent likelihood with average relative error within the tolerance error ($\epsilon$)

# F ABLATION OF DIFFERENT $\epsilon, \delta$ CHOICES FOR NUMERATOR AND DENOMINATOR VERIFIED APPROXIMATION

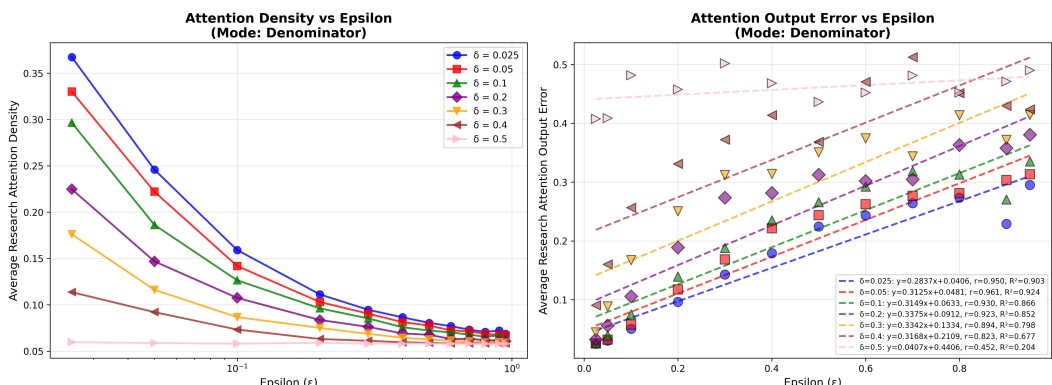

Figure 16: Denominator-verified approximation. Average density and average layer error for different configurations. Note that for reasonable choices of $\delta$, the correlations between user defined $\epsilon$ and average layer error is very high, implying fine-grained control over errors. These plots are created for niah_multikey_2 with $f_s = f_l = 128$, $f_t = 0.05$, $f_b = 0.05$ and we do not lower cap the computed budget by base sampling budget as we do in experiments to produce these plots.

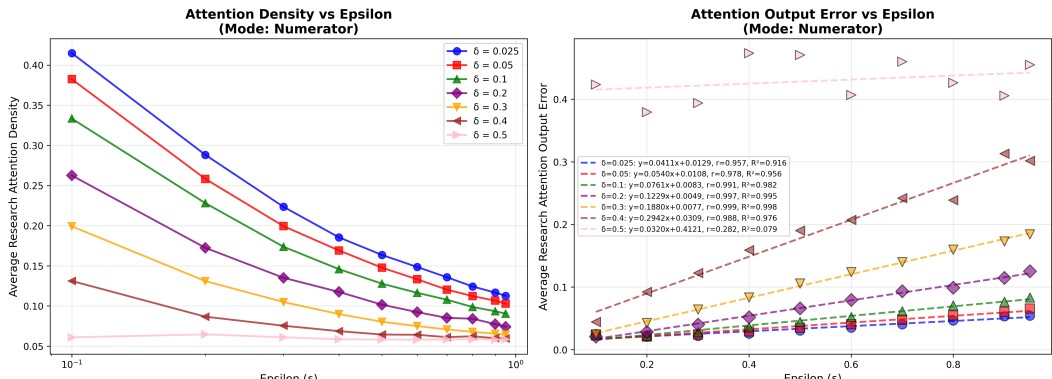

Figure 17: Numerator-verified approximation. Average density and average layer error for different configurations. Note that for reasonable choices of $\delta$, the correlations between user defined $\epsilon$ and average layer error is very high, implying fine-grained control over errors. These plots are created for niah_multikey_2 with $f_s = f_l = 128$, $f_t = 0.05$, $f_b = 0.05$ and we do not lower cap the computed budget by base sampling budget as we do in experiments to produce these plots.

Few things to note in Figure 16 and Figure 17.

- The correlations of average layer error with $\epsilon$ in both cases is very high (almost a linear relation). Which means that both the verified recipes are effective in providing fine-grained control over actual errors.

- Varying the $\epsilon, \delta$ you can span a wide range of sparsity. The $\epsilon$ settings for the numerator have to be higher since numerator operates the guarantee in a higher-dimensional space ($head\_dim$). And in higher dimensions, the volume contained in $\epsilon$ radius ball is exponentially smaller than in lower dimensions ( e.g. 1 for the denominator)

# G  BOOT STRAPPING THE $\sigma^2$ FOR DENOMINATOR AND $\mathrm{TR}(\Sigma)$ FOR NUMERATOR. HOW BIG BASE SAMPLES DO WE NEED?

Table 11 shows the errors in estimating the required statistics for numerator and denominator verified recipes.

Table 11: The average error in estimating variance in the denominator ($\sigma^2$) and trace in numerator ($Tr(\Sigma)$). We see that even with tiny samples, the important variances and traces are approximated very well. The relative error in estimation increases when we look at smaller variances, but these not not very important to estimate since the true budget associated with those variances is orders of magnitude smaller and is lower-capped by base budget in most cases in the implementation.

| niah_multikey_2 | | | |
|---|---|---|---|
| | | | |
| **base sampling rate** | **~Tokens** | **denominator var ( var >0.001)** | **numerator trace ( trace >0.01)** |
| 0.025 | 1000 | 4.74% | 2.77% |
| 0.05 | 2000 | 4.45% | 3.16% |
| 0.1 | 4000 | 3.10% | 2.00% |
| qa_1 | | | |
| **base sampling rate** | **~Tokens** | **denominator var ( var >0.001)** | **numerator trace ( trace >0.01)** |
| 0.025 | 820 | 4.91% | 2.67% |
| 0.05 | 1640 | 3.78% | 2.04% |
| 0.1 | 3280 | 2.57% | 1.30% |
| vt | | | |
| **base sampling rate** | **~Tokens** | **denominator var ( var >0.001)** | **numerator trace ( trace >0.01)** |
| 0.025 | 820 | 5.31% | 2.69% |
| 0.05 | 1640 | 3.63% | 1.72% |
| 0.1 | 3280 | 2.46% | 1.42% |

# H  QQ PLOTS FOR DENOMINATOR

. Figure 18 shows that the estimator of denominator is indeed normally distributed following CLT.

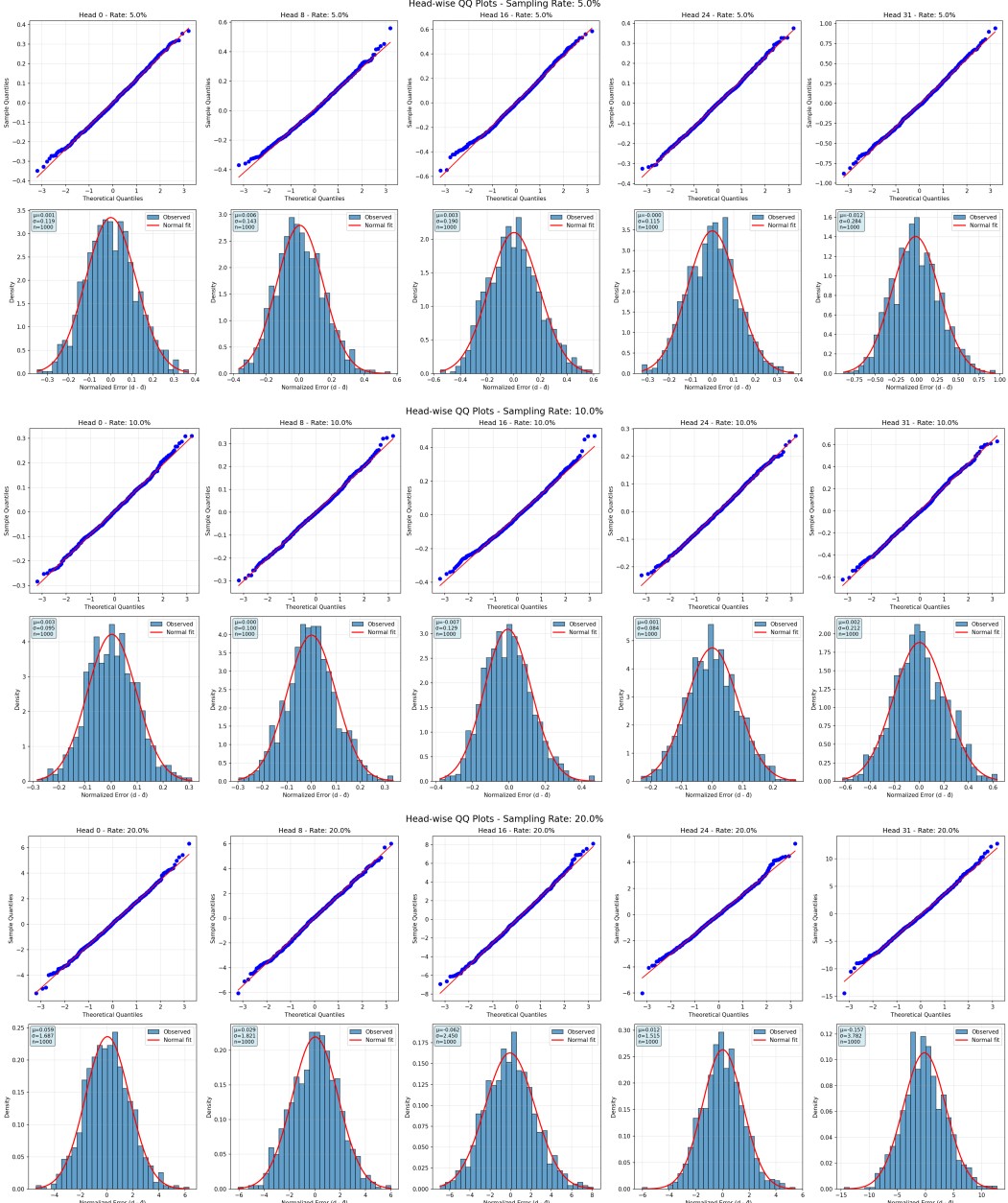

Figure 18: Validity of CLT : The above histogram and QQ plots show that the estimator constructed for the denominator indeed follows a distribution very close to the normal distribution, validating the use of CLT in the vAttention procedure. The sampling rate is relative to context window size which in this experiment is 32K

# I SENSITIVITY ANALYSIS FOR DIFFERENT PARAMETERS OF VATTENTION

To understand the stable region of parameters, we perform the following experiment. Starting from a natural config of

```
sink_size=128,
window_size=128,
HashAttentionTopK(heavy_size=0.05)
base_rate_sampling=0.05,
epsilon=0.05,
delta=0.05,
```

we vary each individual parameter one at a time in the following ranges

```
sink_size=[0, 2, 4, 8, 16, 32, 64, 128]
window_size=[0, 2, 4, 8, 16, 32, 64, 128]
HashAttentionTopK(heavy_size=[0, 0.005, 0.01, 0.025, 0.05, 0.1]
base_rate_sampling=[0, 0.005, 0.01, 0.025, 0.05, 0.1]
epsilon=[0.025, 0.05, 0.1, 0.2, 0.3, 0.4, 0.5]
delta=[0.025, 0.05, 0.1, 0.2, 0.3, 0.4, 0.5]
```

The layer errors observed are presented in the following figure.

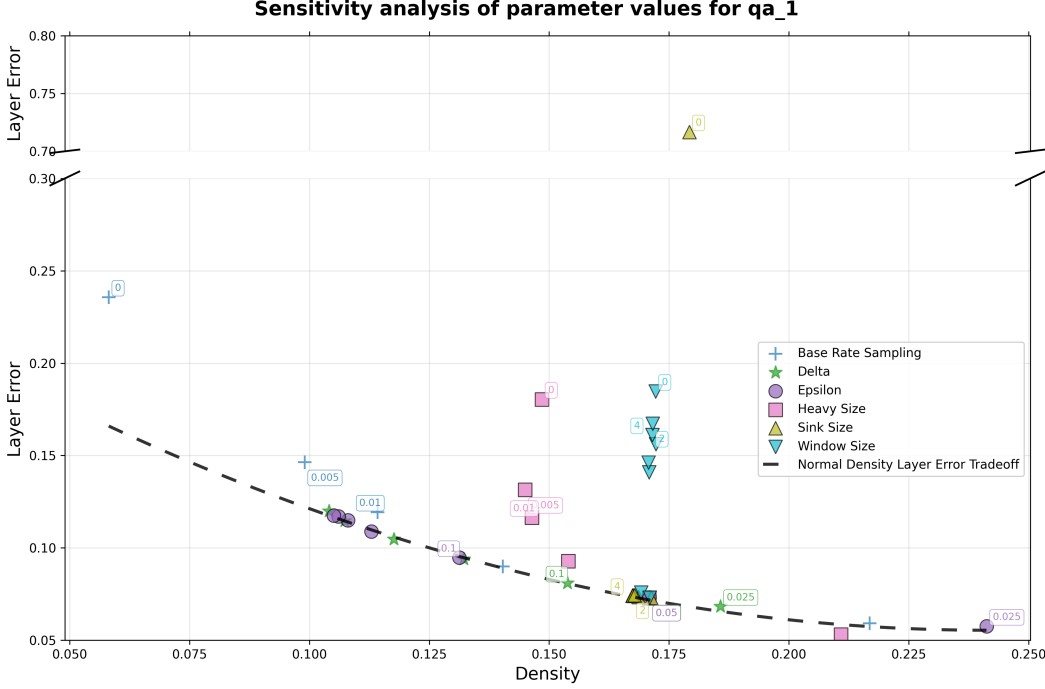

Figure 19: The figure shows layer error as a function of density while varying each parameter across a range of values mentioned in manuscript. The black fitted curve represents the typical relationship between layer error and density. Different parameter settings move the system along this curve, and departures from it reveal unstable regions for each parameter. From these trends we can conclude that sink size and window size should not be extremely small. Setting either of them to zero leads to very large errors. Sink sizes of at least two and window sizes of at least sixty four remain stable. The base rate for sampling should also not be too small. A value of at least 0.025, meaning 2.5% of the context window, is stable. The heavy size for top k should again not be too small, and values of 0.025 or higher are stable. By choosing safe values for these core parameters, the layer error versus sparsity tradeoff can then be explored through the epsilon and delta values.

## J  WIDER EMPIRICAL RESULTS

| Model | Density | DoubleSparsity | MagicPig | OracleTopK | OracleTopP | PQCache | dense | vAttention(OracleTopK) |
|---|---|---|---|---|---|---|---|---|
| Qwen3-30B-A3B-Instruct-2507 | 2% | 22.00 | 20.67 | 90.58 | 92.27 | 90.91 | - | 90.89 |
| | 5% | 25.22 | 32.50 | 91.11 | 91.75 | 90.94 | - | 90.89 |
| | 10% | 36.20 | 38.43 | 91.41 | 91.33 | 91.52 | - | 90.80 |
| | 20% | 59.57 | 69.25 | 91.08 | 91.33 | 91.17 | - | 90.63 |
| | 100% | - | - | - | - | - | 91.02 | - |
| Qwen3-4B-Instruct-2507 | 2% | 18.39 | 18.72 | 86.33 | 88.83 | 85.80 | - | 87.61 |
| | 5% | 21.55 | 29.33 | 87.44 | 88.06 | 87.56 | - | 88.17 |
| | 10% | 26.02 | 39.63 | 87.61 | 88.33 | 87.67 | - | 86.94 |
| | 20% | 47.50 | 76.37 | 88.22 | 87.72 | 87.67 | - | 88.22 |
| | 100% | - | - | - | - | - | 88.67 | - |
| Llama-3.1-8B-Instruct | 2% | 34.90 | 16.28 | 74.10 | 86.07 | 68.95 | - | 87.18 |
| | 5% | 52.83 | 24.83 | 83.83 | 87.01 | 83.14 | - | 86.72 |
| | 10% | 74.75 | 30.20 | 86.37 | 87.45 | 86.29 | - | 87.50 |
| | 20% | 83.40 | 44.08 | 86.90 | 87.62 | 86.98 | - | 88.11 |
| | 100% | - | - | - | - | - | 87.89 | - |
| Llama-3.2-1B-Instruct | 2% | 8.50 | 6.05 | 25.38 | 31.30 | 21.18 | - | 36.78 |
| | 5% | 12.12 | 11.80 | 32.66 | 34.74 | 30.47 | - | 37.80 |
| | 10% | 15.57 | 11.80 | 35.22 | 35.81 | 36.18 | - | 37.83 |
| | 20% | 21.81 | 15.78 | 35.89 | 36.00 | 35.97 | - | 37.51 |
| | 100% | - | - | - | - | - | 37.47 | - |
| Llama-3.2-3B-Instruct | 2% | 18.57 | 17.22 | 41.57 | 51.19 | 39.30 | - | 59.25 |
| | 5% | 24.32 | 21.86 | 47.32 | 56.41 | 46.32 | - | 65.45 |
| | 10% | 30.77 | 21.86 | 53.33 | 59.34 | 51.72 | - | 65.09 |
| | 20% | 40.26 | 36.19 | 59.00 | 62.15 | 59.25 | - | 65.95 |
| | 100% | - | - | - | - | - | 66.25 | - |

Table 12: Model performance across different baselines and sparsity levels

