# OpenReview forum: "vAttention: Verified Sparse Attention via Sampling"
_ICLR.cc/2026/Conference — ICLR 2026 Poster_

### Official Review · Reviewer_rduG · 2025-10-24

**Soundness:** 3
**Presentation:** 3
**Contribution:** 2
**Rating:** 2
**Confidence:** 5

**Summary:**

This paper combines top-k attention with a token sampling approach, seeking an interpolation of the two. It seeks provable and tunable approximation guarantees of the attention function and a practical method that can be deployed to save time and memory without sacrificing model quality. Numerous experiments are performed against many benchmarks to validate the method's performance.

**Strengths:**

1. The paper is well-written, with the exception of some details. It is concise, to the point and effective at communicating its message.
2. The paper tackles the important problem of achieving efficient attention in Transformers without sacrificing model quality. Improvements in this space undoubtedly have profound consequences on the landscape of AI. In this space, the paper contributes a method that offers a lot of practical promise by combining the existing approaches of top-k and sampling attention.
3. The paper performs numerous experiments on a variety of benchmarks to support its claims. There are also lots of ablation studies with other attention methods.
4. The paper quantifies the tradeoff between approximation quality and model decline, at least empirically. This is something very few works in this space do, so it is an important contribution by itself.

**Weaknesses:**

1. The idea of combining sampling and top-k attention is not novel to this paper. The work of [1], for instance, seems to precisely propose the vAttention estimator in eq. (5). Furthermore, [1] also analyzes the approximation guarantees of their estimator rigorously. Beyond [1], numerous other works rigorously analyze the approximation quality of subquadratic attention mechanisms [2,3,4], making me feel uneasy about this paper's claim to  be the "first" algorithm to rigorously allow for quality-efficiency tradeoff control. I would say that the point of departure of this work from [1] and other such works seems to be mainly the fact that this work makes the sample size a tunable hyperparameter, and I worry that this is not a novel enough contribution.
    * That being said, the experimental and empirical study provided by this paper are another one of its contributions. Prior works have not analyzed top-k attention in such an extent, so these insights are definitely valuable to the community. However, the paper is suggesting that it is the first to propose these methods and rigorously analyze them, which, in my opinion, is not accurate.
2. The mathematical rigor of the paper has some notable issues:
    * Lemma 4.1: The $\Phi$ function in Lines 301-303 is applied to the entire term? It is a little hard to see what's happening here.
    * Lemma 4.1: Why are $r_i$ considered random variables with some covariance matrix $\Sigma$? The distribution on the scores is highly unknown and $\Sigma$ is impossible to estimate. Yet the lower bound on $b$ depends on the trace of $\Sigma$. The authors mention this and ultimately set the threshold arbitrarily, but the formal algorithm cannot be stated in terms of this $\Sigma$.
    * Lemma 4.1: The use of CLT here is a bit troublesome. The argument can only work in the limit $n_s \to \infty$, in which case I am also confused as to how the lower bound is ultimately proven. Hoeffding's inequality can salvage things and give a concrete bound, but the paper does not have this proof. Instead it is claimed that in practice the CLT-bound is superior, which again raises the question of how $\Sigma$ is calculated. As a whole, Lemma 4.1 is a bit weak in my opinion.
    * Lemma 4.2: This is the approximation guarantee. I don't think it parses very well: The probability is multiplied with $||N/D||_2$? Also, the statement is shown for $||N||_2/D$ instead in the appendix.

Overall, the paper makes a compelling case for the use of top-k and sampling attention (or interpolation of these) in practice. However, I feel like it currently suffers from issues of originality and rigor.

[1] Haris, Themistoklis. "kNN Attention Demystified: A Theoretical Exploration for Scalable Transformers." The Thirteenth International Conference on Learning Representations.
[2] Han, I., Jayaram, R., Karbasi, A., Mirrokni, V., Woodruff, D. P., & Zandieh, A. (2023). Hyperattention: Long-context attention in near-linear time. arXiv preprint arXiv:2310.05869.
[3] Choromanski, K., Likhosherstov, V., Dohan, D., Song, X., Gane, A., Sarlos, T., Hawkins, P., Davis, J., Mohiuddin, A., Kaiser, L. and Belanger, D., 2020. Rethinking attention with performers. arXiv preprint arXiv:2009.14794.
[4] Alman, Josh, and Zhao Song. "Fast rope attention: Combining the polynomial method and fast fourier transform." arXiv preprint arXiv:2505.11892 (2025).

**Questions:**

1. For the experimental section, it is mentioned that it is expensive to calculate the numerator so it is skipped? Why is it expensive? How can we afford to skip that approximation without suffering later on? This seems to be a different algorithm than what was argued earlier.

---

> ### Author Response · Authors · 2025-11-21
> **vAttention vs. “kNN Attention Demystified” and [2,3,4] [1/2]**
>
> We thank the reviewer for their time and effort for evaluating our submission
>
>
> ### **vAttention vs. “kNN Attention Demystified”**
>
>
>
> **Summary of the paper in relation to vAttention:** This paper develops a subquadratic attention mechanism by reformulating self-attention as an expectation under the softmax distribution, made tractable through lazy Gumbel sampling. It provides a procedure for approximating gradients and presents preliminary evidence on the trainability of k-NN Attention. Although the central proposal relies on applying Gumbel sampling to top-k tokens together with random sampling over the remaining tokens (implemented via approximate top-k rather than oracle top-k), Section 2.3 diverges from this main approach and introduces a “simpler algorithm,” which corresponds to the idea of combining top-k and sampling. The authors analyze this formulation in Theorem 8 (which originates from Mussmann (2017).) , giving $\epsilon, \delta$ bounds for each coordinate of the SDPA output, and note that this is the version used in their experiments.
>
> While, vAttention’s idea of combining top-k with sampling is partially explored in this paper, we argue that the main contribution of vAttention is much more than equation (5) including the target problem space, simple and clean theory which can be converted into a practical algorithm and extensive LLM experiments at practical scale with impactful results. We elaborate on each of these points contrasting them against kNN Attention paper, below
>
> 1. **Problem scope and approach.** First, the problem settings of the two papers differ substantially. kNN-Attention aims to develop a subquadratic alternative to softmax attention for training models from scratch. In contrast, vAttention is explicitly designed to approximate softmax attention at inference time, with the goal of accelerating long-context inference for existing open-source LLMs. This distinction matters because it drives different choices of matrix shapes and analytical assumptions: the kNN-Attention analysis is tailored to the prefill scenario, whereas vAttention’s analysis focuses on the decoding scenario. As we show below, this difference has important implications for both the theory and its practical relevance.
>
> 2. **Algorithm from theory:** We attempted to derive a practical algorithm from the theoretical guarantees in Theorem 8. However, when applied to our RULER experiments with a 32K context length, the theorem’s prescribed values for (k,l) (in  kNN attention’s notation k is the top-k and l the sample size) seem impractically large. Even after ignoring the hidden constants in the big-O terms—which could themselves be substantial—the resulting token counts required for reasonably small $\epsilon$ and $\delta$ are considerably large. Also note that this $\epsilon$ guarantee is per dimension. So possibly these numbers are missing a factor of $\sqrt{d}$ where $d$ is head dimension.
>
> | $\delta \rightarrow$  |       0.01 |      0.05 |       0.1 |
> |-----------------------|-----------:|----------:|----------:|
> | $\epsilon \downarrow$ ||||
> | 0.01 | 1007936.84 | 450763.06 | 318737.62 |
> |0.05 |  201587.37 |  90152.61 |  63747.52 |
> |0.1 |  100793.68 |  45076.31 |  31873.76 |
>
> It is still possible, in principle, to derive an algorithm from Theorem 8 by carefully accounting for the hidden constants and relaxing the $\epsilon$ and $\delta$ parameters. However, since the kNN-Attention paper does not focus on this algorithm, exploring it would require deeper theoretical analysis. One clear distinguishing factor in  vAttention’s theory from Theorem 8 is that it is simple and adaptive to the data at hand, whereas theorem 8 is data agnostic, potentially causing larger budgets predictions.
>
> Going further into the theory of knn-attention and the theoretical papers it builds upon is out of the scope of this rebuttal, but as it stands the theory is not specifically tuned for the problem that is of interest to vAttention.
>
> 3. **Actually improving off-the-shelf LLM inference via sparse attention**: While the kNN-Attention approach could, in principle, improve inference for off-the-shelf LLMs, this remains purely speculative. Its experiments are limited, showing only training loss evolution in two toy settings. In contrast, vAttention demonstrates practical impact with extensive experiments on state-of-the-art open-source LLMs and popular long-context datasets, clearly validating its effectiveness for improving inference. In fact, we have extended the empirical evaluation to include 5 SOTA models (1B sized to 30B sized) including dense and moe architectures, and compared against a total of 6 SOTA sparse attention baselines (DoubleSparsity, HashAttention, MagicPig, OracleTopK,OracleTopP,PQCache)

---

> ### Author Response · Authors · 2025-11-21
> **vAttention vs. “kNN Attention Demystified” and [2,3,4] [2/2]**
>
> We thank the reviewer for bringing this paper to our attention. We were not aware of this work. We will ensure its relevance and precedence is clearly acknowledged in our work. We respectfully urge the reviewer to reconsider the unique value added by vAttention in making the combination of top-k selection and sampling both practical and impactful. If reviewer concurs that the idea of combination of top-k and sampling is indeed promising, it is more likely to reach and benefit the community through vAttention than through the kNN-Attention paper in its current state, which is primarily theoretical, targets a different problem and where this observation is buried within a brief 14-line section amid the Gumbel sampling proposal, gradient approximation theory, and minimal training experiments.
>
>
> ### **Comparing vAttention to other papers (citations 2,3,4)** :
>
> While **Performers** focus on linearizing attention training via kernel approximations and **Fast RoPE Attention** addresses the theoretical complexity of rotating embeddings using FFTs , **vAttention** specifically targets inference latency by introducing a "verified" sparse mechanism. Unlike **HyperAttention**, which relies on Locality Sensitive Hashing (SortLSH) and spectral bounds for heavy-hitter discovery, **vAttention** is method-agnostic regarding heavy-hitters and uniquely employs statistical bounds (CLT/Hoeffding) to dynamically calculate the exact sample budget needed to meet a user-defined error tolerance $(\epsilon, \delta)$. This allows **vAttention** to offer granular, reliability-focused control over the quality-efficiency trade-off during decoding, distinct from the fixed asymptotic or kernel-based guarantees of prior work.

---

> ### Author Response · Authors · 2025-11-21
> **Relaxation of Theoretical Guarantees: (Q1)**
>
> ### **Relaxation of Theoretical Guarantees: (Q1)**
>
> We realized that the way denominator-, numerator-, and verified-SDPA (per-head) recipes were distinguished in our writing incorrectly suggesting that one is a relaxation of another. This was not the intention. We will clarify the presentation and upload the revised version by Dec 3 to avoid any confusion.
>
> Denominator, numerator, and per-head attention, the three granularities discussed in the paper, are all intermediate computations affected by sparse approximation. For each of these, we propose a corresponding verified recipe. In general, verified-X means that we can approximately compute X within user-specified $(\epsilon, \delta)$ guarantees. The guarantees are specific to computation X (e.g. denominator, numerator, per-head SDPA etc). They do not make a statement on any larger computation encapsulating X such as per-layer-attention output or entire model output. Thus verified-denominator, verified-numerator and verified-SDPA (per-head) are all independent algorithms guaranteeing specific computations. One is not intended to be a relaxation of the other.
>
> Given that we have three choices of verified-X, the selection of X depends on two factors
> - Control: How interpretable and fine-grained the control is that verified-X provides over the quality of larger computation encapsulating X. We use per-layer attention error as a larger computation.
> - Cost: How expensive the corresponding budget computation is under each verified-X option
>
> As shown in Figure 1 (right), and in the newly added more extensive plots for both numerator and denominator in Appendix F (Figures 16 and 17), numerator and denominator offer similarly fine-grained control over model errors. The correlation between the layer-level attention output error (i.e., the error of the attention module after combining all heads and applying the O projection) and the user-specified $\epsilon$ remains very high for both. Thus, both are strong candidates. We prefer verified-denominator because it is simpler and requires access only to the base sample of keys, whereas verified-numerator and verified per-head attention require access to both keys and values, effectively doubling the memory needed for the base sample.
>
> **On related “Why is [numerator computation] expensive?”**
>
> Computing the budget using the verified numerator and verified denominator requires calculating different statistics ($Tr(\Sigma)$ and $||N||$ for the numerator, and $\sigma^2$ and $D$ for the denominator). There are three considerations regarding cost:
> 1. Computational complexity of estimating from samples: Numerator computations occur in the head dimension and use both K and V caches, making them inherently expensive.
> 2. Sample complexity: This refers to the sample size required for accurate estimation. Theoretically, the sample complexity for the numerator can be higher or lower than that for the denominator, depending on the actual distribution. In practice, we find the sample complexity to be similar. See the table below for estimation errors of the numerator and denominator.
> 3. Memory footprint of samples: Numerator computations require both K and V caches, whereas denominator computations require only K. As a result, the memory footprint for numerator computations is roughly twice that of the denominator for the same sample complexity.

---

> ### Author Response · Authors · 2025-11-21
> **Other Questions**
>
> ### **Other Questions / conerns**
>
> **Lines 301-303 application of $\Phi^{-1}$**
>
>  The $\Phi^{-1}$ function is applied to $( 1 - \delta/2)$. More details on how we arrive at this expression are present in appendix D.
>
> **Lemma 4.1 analysis assumes knowing $\Sigma$ and then estimates are used.**
>
> Estimating unknown statistics and using them is a standard procedure in developing statistical algorithms. This is popularly referred to as plug-in estimators. Some references that talk about this are “All of Statistics, Wasserman (Chapter 7)”, “An Introduction to Bootstrap” , Efron and Tibshirani.
> To show how well we approximate meaningful statistics with the sample sizes that we choose, we show the results below. We find that even with small samples we are able to approximate the statistics with very small errors. This is expected since these estimators have fast convergence.
>
>
>
> |   **niah_multikey_2**  |         |  |   |
> |:------------------:|:-------:|:------------------------------:|:-------------------------------:|
> |  |   |  |         |
> | base sampling rate | ~Tokens | denominator var ( var > 0.001) | numerator trace ( trace > 0.01) |
> | 0.025 |    1000 |4.74% | 2.77% |
> |  0.05 |    2000 |4.45% | 3.16% |
> |   0.1 |    4000 |3.10% | 2.00% |
> |        **qa_1**        |         |      |       |
> | base sampling rate | ~Tokens | denominator var ( var > 0.001) | numerator trace ( trace > 0.01) |
> | 0.025 |     820 |4.91% | 2.67% |
> |  0.05 |    1640 |3.78% | 2.04% |
> |   0.1 |    3280 |2.57% | 1.30% |
> |         **vt**         |         |      |       |
> | base sampling rate | ~Tokens | denominator var ( var > 0.001) | numerator trace ( trace > 0.01) |
> | 0.025 |     820 |5.31% | 2.69% |
> |  0.05 |    1640 |3.63% | 1.72% |
> |   0.1 |    3280 |2.46% | 1.42% |
>
> **Lemma 4.1: The use of CLT /  Hoeffding.**
>
> The proposal of vAttention can work with both CLT and Hoeffding. In fact, we do analysis on how computed budgets look like with both of these methods in appendix E. We choose CLT in the main paper since firstly, we think CLT offers easier exposition and CLT reasonably approximates our scenario at hand.
> The target regime of the problem is a long context where reasonable budgets also provide us with a large sample size. For example at 128K or 1M context size (the real application of sparse attention) will give us sample sizes of 12K or 100K at 10% sparsity.  Such large sample sizes are good enough for application of CLT. We can see this in practice ( See QQ plots for the estimator in newly added Appendix H) and also in theory. For example, In case of denominator estimation, all the moments of the estimator are bounded (specifically the 3rd moment). Thus, we can apply Berry-Esseen Theorem to know that the distribution converges at the rate of $1/\sqrt{n}$.
>
>
> **Lemma 4.2: Typo**
> Thanks for pointing out the typo in the equation. The right bracket is misplaced. We have corrected the manuscript.
>
>
> Please let us know if we missed any of the concerns / questions or if there are additional questions. We are happy to clarify further.

---

> > ### Comment · Reviewer_rduG · 2025-11-23
> > **Response to authors**
> >
> > Hello,
> >
> > Thank you for responding extensively to my comments.
> >
> > I do agree that the paper' contribution does indeed go beyond the work of [Haris; 25] because it clearly showcases the practicality of such an approach and performs rigorous experiments on LLM inference. As the authors noted, that prior work seems to offer more of a theoretical insight on top-k attention, where as vAttention gives a practical and implementable algorithm. It would still be valuable, I think, to position the paper accordingly within the context of that prior work as well. Its theoretical insights might be able to strengthen the paper's claims further.
> >
> > Beyond that, I appreciate the clarifications on the theory. However, I am still a little skeptical about the theoretical insights the paper offers and I worry they could confuse or mislead the reader:
> > * Lemma 4.1 simply gives a CLT-based approximation for sampling a vector-sum. Though CLT can morally be used to give an estimate, it is only valid in the limit and thus it is a bit unclear what the $b$ quantity is in practice. It also depends a lot on an accurate estimate of $\Sigma$ which is done using only $f_b$ samples. To be correct it feels like the theory should incorporate an error in the estimation of the variance as well as an error in the approximation of CLT in the limit.
> > * Lemma 4.2 is the statement that to approximate a fraction it suffices to approximate the numerator and denominator. This is fairly standard.
> > * Theorem 4.3 combines these claims.
> >
> > Additionally, it does feel like the theory is disregarded a little bit in ways that are not fully specified. The paper makes some valid algorithmic "relaxations" that help boost the algorithm's performance, but if it breaks from its narrative too much then it causes confusion. Ultimately the algorithm presented should be the algorithm used. I think the theoretical insights may be helpful if presented more as intuitive guidelines rather than explicit guarantees, especially since the primary strength of the paper is, to me, its solid practicality, GPU-aware implementation and combination of various attention approximation methods.
> >
> > I would love to see these concerns addressed further via a revision. As it stands, though my score is increased, I maintain some skepticism towards acceptance.

---

> > > ### Author Response · Authors · 2025-11-28
> > > **Response to the reviewer**
> > >
> > > We appreciate the reviewer updating their score and confidence. We have updated the manuscript to reflect the writing changes promised for "Relaxation of Theoretical Guarantees". Please find our comments on the response.
> > >
> > > “It would still be valuable, I think, to position the paper accordingly within the context of that prior work as well.”
> > >
> > > While previous approaches offer related perspectives, these approaches were neither designed nor evaluated for the specific problem of inference time acceleration of off-the-shelf LLM models . We agree that they are important but exploring them for this problem is an independent undertaking.
> > >
> > > “CLT can morally be used to give an estimate, it is only valid in the limit”
> > >
> > > As shown in our new results in Appendix I, the distributions we treat as approximately normal (via the Central Limit Theorem) are indeed extremely close to normal, validating this approximation. More importantly, the method performs well empirically, as demonstrated by our comprehensive evaluation.
> > >
> > > "It also depends a lot on an accurate estimate of $\Sigma$ which is done using only $f_b$ samples"
> > > We only need to estimate $Tr(\Sigma)$ which is a scalar and much easier to estimate than $\Sigma$ which is $d \times d$ matrix.
> > >
> > > “To be correct it feels like the theory should incorporate an error in the estimation of the variance as well as an error in the approximation of CLT in the limit.”
> > >
> > > While additional theoretical refinements—such as incorporating estimation error and CLT convergence rates—are certainly possible, doing so is a standard, even if nontrivial, extension. We plan to continue deepening the theoretical foundations of vAttention as this line of work progresses.
> > >
> > > “it does feel like the theory is disregarded a little bit in ways that are not fully specified”
> > >
> > > We agree that the presentation of the verified-X algorithms in the paper was confusing. We have clarified this extensively in our rebuttal above and have updated the manuscript accordingly to remove any confusion. The theory is strongly tied to how these algorithms are implemented.

---

### Official Review · Reviewer_YGRV · 2025-10-28

**Soundness:** 3
**Presentation:** 3
**Contribution:** 3
**Rating:** 6
**Confidence:** 3

**Summary:**

This paper introduces vAttention, a sparse attention mechanism that unifies deterministic top-k selection with sampling-based estimation and provides formal ($\epsilon, \delta$)
guarantees on the approximation of full attention. Theoretical results show that the estimator achieves bounded error under Central Limit Theorem–based sampling, and experiments across Llama-3.1, DeepSeek, and Mistral models demonstrate strong quality-efficiency trade-offs, often outperforming existing top-k methods.

**Strengths:**

1. The mathematical derivation is clear and connects attention approximation to classical sum-estimation theory.

2. Shows consistent empirical gains on long-context benchmarks and stable long-generation quality.

3. The framework is compatible with existing top-k implementations.

4. Writing and organization are clear, with well-motivated theoretical and experimental sections.

**Weaknesses:**

1. No ablation on the relaxation that only approximates the denominator.

2. No GPU runtime results or CUDA implementation to verify efficiency gains.

3. Missing quantitative reporting on selected token counts and achieved sparsity.

4. Comparison to oracle top-p and newer top-p-based methods is incomplete.

5. Parameter selection procedure for $\delta$ and $\epsilon$ is heuristic and underexplained.

**Questions:**

This paper is solidly motivated and theoretically sound, and the idea of providing formal guarantees for sparse attention is timely. However, several issues limit its empirical completeness and clarity. Below, I describe these concerns and follow with specific technical questions.

The relaxation that approximates only the denominator is a major simplification. While it is argued to control bias and match observed errors, there is no ablation comparing the full
 $(\epsilon, \delta)$ approximation to this relaxed version. The paper would benefit from quantifying how much quality or efficiency is gained or lost through this relaxation. In addition, since the method introduces user-defined $(\epsilon, \delta)$ parameters, it would be useful to know how these affect token selection and runtime. The paper does not clearly show how many tokens are chosen for a given accuracy on the benchmark compared to baselines.

The efficiency evaluation focuses on CPU-bound experiments and does not provide GPU results. Sparse attention acceleration is most relevant for GPU decoding workloads, so the lack of CUDA implementation or end-to-end throughput measurements makes it hard to judge the real benefit. Even a partial GPU prototype would help demonstrate scalability.

The comparison with oracle top-p is not conclusive. In some plots (Appendix A.1), vAttention performs worse than oracle top-p at the same density, but this is not discussed. Since top-p offers a different adaptive coverage mechanism, further comparison or integration could clarify the relative advantages.

Finally, parameter tuning remains underexplained. The authors mention grid search over $(\epsilon, \delta)$ and sampling fractions but give no practical guideline for setting them. For deployment, it would be important to know if there exists a stable range that generalizes across models or if these need per-benchmark tuning.


Questions:

1. How large is the empirical performance gap between the full numerator-denominator estimation and the denominator-only relaxation?

2. In Eq. (5), the deterministic and stochastic terms are directly summed. Would importance weighting or probability normalization further reduce bias?

3. Are there engineering challenges that prevent implementing the methods in CUDA? Could it reuse GPU primitives for top-k?

4. What are the average token counts or densities achieved under different $(\epsilon, \delta)$ values, and how do they translate to runtime gains on GPU?

5. When vAttention underperforms oracle top-p at fixed sparsity, is that due to attention distribution skewness, or to the approximation of only the denominator? Could vAttention be compatible with existing top-p methods?

6. Why does vAttention sometimes outperform dense attention on AIME (Table 2)?

---

> ### Author Response · Authors · 2025-11-21
> **Response to reviewer YGRV [1/n]**
>
> We thank the reviewer for their time and effort for evaluating our submission
>
>
>
> ### **Relaxation of Theoretical Guarantees**:
>
> We realize that our presentation of the denominator-, numerator-, and verified-SDPA (per-head) recipes may have incorrectly suggested that one is a relaxation of another. This was unintended. We will clarify this distinction and upload the revised version by Dec 3.
>
> Denominator, numerator, and per-head attention, the three granularities discussed in the paper, are distinct intermediate computations affected by sparse approximation. For each, we propose a corresponding verified recipe. In general, verified-X means that we can approximately compute X within user-specified $(\epsilon, \delta)$ guarantees. These guarantees are specific to computation X (e.g. denominator, numerator, per-head SDPA etc) and do not directly bound larger encapsulating computations, such as per-layer attention or model output. Thus verified-denominator, verified-numerator and verified-SDPA (per-head) are independent algorithms guaranteeing specific computations, not relaxations of one another.
>
> Given these three options, the selection of X depends on two factors
> - Control: How interpretable and fine-grained the control is that verified-X provides over the quality of larger computation encapsulating X. We use per-layer attention error as a larger computation.
> - Cost: How expensive the corresponding budget computation is under each verified-X option
>
> As shown in Figure 1 (right), and in the newly added more extensive plots for both numerator and denominator in Appendix F (Figures 16 and 17), numerator and denominator offer similarly fine-grained control over model errors. The correlation between the layer-level attention output error (i.e., the error of the attention module after combining all heads and applying the O projection) and the user-specified $\epsilon$ remains very high for both. Thus, both are strong candidates. We prefer verified-denominator because it is simpler and requires access only to the base sample of keys, whereas verified-numerator and verified per-head attention require access to both keys and values, effectively doubling the memory needed for the base sample.
>
>
>
> **On related note “quantifying how much quality or efficiency is gained or lost”**
>
> The main difference is how we compute budgets in different recipes. The expense comes from sample size needed for estimating these statistics accurately since we need to either store this sample on GPU or fetch it for budget computation. The computation itself, especially in the context of offloaded KV Cache, is not the bottleneck. The sample size needed for estimating numerator statistics  can be more or less than that required for estimating denominators depending on how correlated the different dimensions are. We did an empirical study on the estimation quality at different sample sizes and datasets and here are the results. With small sample sizes, we are able to accurately determine the statistics required for both numerator and denominator. So practically there is not much difference in using one over the other.
>
>
>
>
>
>
>
> |   niah_multikey_2  |         |                                |                                 |
> |:------------------:|:-------:|:------------------------------:|:-------------------------------:|
> |                    |         |                                |                                 |
> | base sampling rate | ~Tokens | denominator var ( var > 0.001) | numerator trace ( trace > 0.01) |
> |              0.025 |    1000 |                          4.74% |                           2.77% |
> |               0.05 |    2000 |                          4.45% |                           3.16% |
> |                0.1 |    4000 |                          3.10% |                           2.00% |
> |        **qa_1**        |         |                                |                                 |
> | base sampling rate | ~Tokens | denominator var ( var > 0.001) | numerator trace ( trace > 0.01) |
> |              0.025 |     820 |                          4.91% |                           2.67% |
> |               0.05 |    1640 |                          3.78% |                           2.04% |
> |                0.1 |    3280 |                          2.57% |                           1.30% |
> |         **vt**         |         |                                |                                 |
> | base sampling rate | ~Tokens | denominator var ( var > 0.001) | numerator trace ( trace > 0.01) |
> |              0.025 |     820 |                          5.31% |                           2.69% |
> |               0.05 |    1640 |                          3.63% |                           1.72% |
> |                0.1 |    3280 |                          2.46% |                           1.42% |

---

> ### Author Response · Authors · 2025-11-21
> **Response to reviewer YGRV [2/n]**
>
> ### **Observed sparsity and tokens**
> The tokens are represented as sparsity levels. 10% sparsity/density ( we use sparsity and density interchangeably in the paper and we will fix the usage to density) at 32K means using 3.2K tokens for attention computation. In all tables / data, we specify the achieved sparsity. Did we misunderstand the question / concern?
>
> For RULER HARD dataset, at 10% sparsity / density
> | Dataset | Context Length | Tokens used |
> | :--- | :--- | :--- |
> | qa_1, qa_2, fwe, vt | ~32K | ~3.2K |
> | niah_multikey_2/3 | ~40K | ~4K |
>
>
>
>
> ### **Sensitivity to Hyperparameters:**
>
> Most sparse attention methods, including top-k approaches need some calibration of sorts. Choosing the hyper parameters can be thought of as a calibration step, similar to how we need to create a model specific channel config in Double Sparsity or MLP weights need to be trained for HashAttention. Additionally, we provide easy to use scripts where we can quickly search for the parameters in the specified grid to simplify the process.
>
> Moreover, there is a natural and stable choice for most parameters. In our experiments, we generally use fixed values for $f_s$​ and $f_l$​, and perform a coarse-grained search for the remaining parameters. To demonstrate stability, we conducted an extensive sensitivity analysis, now presented in Appendix I. This analysis confirms our expectation that choosing safe values for $f_s​, f_l​, f_t​, f_b$​ (for example, $f_s >= 2 , f_l >=64, f_t, f_b >= 0.025$ ) places the method on the standard quality–efficiency tradeoff curve, which can then be traversed through appropriate choices of $\epsilon,\delta$.
>
> Natural configurations can be used across the board (specifically for in-the-wild deployment). For instance if we use the same parameters as we used for AIME ($f_s = 128 , f_l =128, f_t, f_b = 0.025, \epsilon=\delta=0.05$) on RULER-HARD, we get the following performance which essentially matches the full attention performance at 10-15% tokens.
>
>
> | Dataset | Density | Quality | Dense |
> | :--- | :--- | :--- | :--- |
> | qa_1 | 0.143 | 80.5 | 80.5 |
> | qa_2 | 0.136 | 51 | 51.5 |
> | niah_multikey_2 | 0.111 | 99.5 | 99.5 |
> | niah_multikey_3 | 0.114 | 100 | 100 |
> | vt | 0.16 | 97.4 | 97.4 |
> | fwe | 0.103 | 93.17 | 93.17 |
>
>
>
>
> ### **Computational Overhead of Budget Calculation /  End-to-end speed evaluation**
>
> The focus of this paper is long-context decoding. Sparse attention provides little value at shorter sequence lengths, but at very long contexts the KV cache can no longer fit on a single GPU. This forces systems to either offload the KV cache to CPU memory or to shard it across GPUs via context parallelism. In case of offloading, the primary bottleneck is CPU-GPU bandwidth for which we show gains. Evaluating sparse-attention methods under context-parallel execution, requires substantial engineering effort and is beyond the scope of this paper.
>
>
> ### **Oracle-top-p and vAttention**
>
> The good performance of oracle-top-p over vattention  is not statistically significant. While oracle-top-p performs marginally better (e.g. In loogle/shortdepqa the max difference in the quality curves is 0.003 points)  in terms of final quality metric of the benchmark/dataset in some settings, the local attention errors, which is much more statistically significant and stable metric,  oracle-top-p is consistently worse than vattention across different benchmark/datasets and sparsity levels.
>
> While writing the paper we did try to add an approximate-top-p baseline of Tactic. However, they do not have an open source implementation and our efforts to implement the baseline did not reproduce the numbers from their paper. If you could direct us to an approximate top-p baseline with good open source code, we are happy to include it in our final paper.
>
> Moreover, vAttention can be combined with any deterministic index selection strategy including oracle-top-p or its approximate variants. We support this seamlessly in our code. However, in our experiments, we do not see much benefit for using this combination. An example of combination is,
>
> | Dataset | Metric | oracle-top-p ($p=0.85$) | oracle-top-p($p=0.75$) + vAttention |
> | :--- | :--- | :--- | :--- |
> | **qa_1** | Quality | 76.5 | 80 |

---

> ### Author Response · Authors · 2025-11-21
> **Response to reviewer YGRV [3/n]**
>
> ## **Other questions**
> >“In Eq. (5), the deterministic and stochastic terms are directly summed. Would importance weighting or probability normalization further reduce bias?”
>
> The equation already has inverse probability weights which ensure that numerator and denominator both are unbiased estimates.
>
>
>
> >“Are there engineering challenges that prevent implementing the methods in CUDA? Could it reuse GPU primitives for top-k?”
>
> Writing an optimized CUDA kernel for budget computation is indeed a substantial undertaking. In fact, even sub components of the algorithm such as top-k on GPU are areas of active research. For instance, [1]
>
> >Why does vAttention sometimes outperform dense attention.
>
> Sparse attention occasionally outperforms dense attention, a pattern noted in several recent sparse attention papers (Desai 2024, Yang 2024, and others). One can argue that sparse attention may remove distractors and thereby improve quality. However, this phenomenon has not been investigated in depth, likely because the gains are neither consistent nor substantial enough to motivate further study.
>
> [1] Xie, X., Luo, Y., Peng, H. and Ding, C., 2024, September. RTop-K: Ultra-Fast Row-Wise Top-K Selection for Neural Network Acceleration on GPUs. In The Thirteenth International Conference on Learning Representations.
>
>
>
> Please let us know if we missed any of the concerns / questions or if there are additional questions. We are happy to clarify futher.

---

### Official Review · Reviewer_5pmM · 2025-10-28

**Soundness:** 3
**Presentation:** 3
**Contribution:** 3
**Rating:** 6
**Confidence:** 2

**Summary:**

This paper introduces **vAttention**, a sparse attention mechanism with verifiable accuracy control. The key observation is that top-k and random sampling are complementary: top-k captures sharp, heavy-tailed score distributions while sampling better covers near-uniform regions. vAttention unifies them: it deterministically selects heavy-hitters (sink tokens, a local window, and approximate top-k) and then uniformly samples from the residual “long tail.” Users set \\((\epsilon,\delta)\\) and the method computes a per-query, per-head sampling budget via CLT-based estimates. Experiments show clear gains over strong baselines and narrow the gap to full attention, matching full-attention quality under high sparsity (≈10–15% density; ≈12% at 32K) in long-generation settings.

**Strengths:**

1. **Principled guarantees.** vAttention is presented as a practical algorithm that exposes user-controllable \\((\epsilon,\delta)\\) error targets. Empirically, the realized approximation error correlates strongly with the user tolerance \\(\epsilon\\) (correlation \\(>0.99\\)).

2. **Robust hybrid design.** The deterministic heavy-hitter stage plus stochastic long-tail sampling is well-motivated. A simple combination of oracle-top-k with random sampling consistently outperforms either component alone.

3. **Strong empirical results.**
   - **Accuracy:** Improves over HashAttention on RULER-HARD by ≈4.5 points (e.g., +4.6 for Llama-3.1-8B and +4.3 for DeepSeek-R1-Distill-Llama-8B).
   - **Oracle comparison:** vAttention + oracle-top-k can outperform oracle-top-p on RULER-32K, suggesting limits of pure top-k/top-p schemes.
   - **Long generation:** Matches full-attention quality on AIME@32K at ≈12% average density; at ≈16K, densities around 10–15% are reported.

**Weaknesses:**

1. **Theory–practice relaxation.** The paper proves joint \\((\epsilon,\delta)\\) guarantees for **both** SDPA numerator \\(N\\) and denominator \\(D\\) (Theorem 4.3), leveraging a composition of bounds (with Lemma 4.2 for separate approximations). Computing the full budget is reported as expensive. Consequently, **all experiments** adopt a relaxation that provides an \\((\epsilon,\delta)\\) guarantee **only for the denominator** \\(D\\). The “verified” claim in practice is therefore supported by the strong empirical correlation \\(>0.99\\) rather than the full \\(N\\)&\\(D\\) guarantee.

2. **Budget/latency overhead under-measured on GPU.** Speedups are shown in memory-bound regimes with KV on CPU and with a naive PyTorch implementation. A CUDA kernel is left to future work. The end-to-end latency trade-off when KV fits in HBM is not fully benchmarked, so the actual GPU-resident benefit remains uncertain.

**Questions:**

1. **Cost of full \\(N\\)&\\(D\\) guarantee:** What is the empirical overhead and final density increase when enforcing the full Theorem 4.3 guarantee versus the denominator-only relaxation? Is the expense dominated by estimating \\(\mathrm{Tr}(\Sigma)\\) for the vector-valued numerator, and can that be approximated efficiently without breaking the guarantee?

2. **Budget partitioning guidance:** The method composes sink/local/top-k (deterministic) with uniform sampling (stochastic). Experiments grid-search the fractions \\((f_s,f_l,f_t)\\). Is there any theoretical prescription for allocating budget between deterministic heavy-hitters and stochastic sampling to minimize total density for a target \\((\epsilon,\delta)\\)?

3. **Early-token behavior:** For very early tokens in long generation, when the residual set size \\(n_s\\) is small, CLT assumptions are weaker. Does Algorithm 2 switch to heavier reliance on the deterministic set \\(\mathcal{I}_f\\) or adaptive rules for minimum base samples to keep the guarantees meaningful in this regime?

---

> ### Author Response · Authors · 2025-11-21
> **Response to reviewer 5pmM [1/n]**
>
> We thank the reviewer for their time and effort for evaluating our submission
>
>
> ### **Relaxation of Theoretical Guarantees / Cost of computing full guarantee:**
> We realized that the way denominator-, numerator-, and verified-SDPA (per-head) recipes were distinguished in our writing incorrectly suggesting that one is a relaxation of another. This was not the intention. We will clarify the presentation and upload the revised version by Dec 3 to avoid any confusion.
>
> Denominator, numerator, and per-head attention, the three granularities discussed in the paper, are all intermediate computations affected by sparse approximation. For each of these, we propose a corresponding verified recipe. In general, verified-X means that we can approximately compute X within user-specified $(\epsilon, \delta)$ guarantees. The guarantees are specific to computation X (e.g. denominator, numerator, per-head SDPA etc). They do not make a statement on any larger computation encapsulating X such as per-layer-attention output or entire model output. Thus verified-denominator, verified-numerator and verified-SDPA (per-head) are all independent algorithms guaranteeing specific computations. One is not intended to be a relaxation of the other.
>
> Given that we have three choices of verified-X, the selection of X depends on two factors
>
> 1. How interpretable and fine-grained the control is that verified-X provides over the quality of larger computation encapsulating X. We use per-layer attention error as a larger computation.
> 2. How expensive the corresponding budget computation is under each verified-X option
>
> As shown in Figure 1 (right), and in the newly added more extensive plots for both numerator and denominator in Appendix F (Figures 16 and 17), numerator and denominator offer similarly fine-grained control over model errors. The correlation between the layer-level attention output error (i.e., the error of the attention module after combining all heads and applying the O projection) and the user-specified $\epsilon$ remains very high for both. Thus, both are strong candidates. We prefer verified-denominator because it is simpler and requires access only to the base sample of keys, whereas verified-numerator and verified per-head attention require access to both keys and values, effectively doubling the memory needed for the base sample.
>
>
> **On a related note, “Is the expense dominated by estimating  for the vector-valued numerator,”?**
> Computing the budget using the verified numerator and verified denominator requires calculating different statistics ($Tr(\Sigma)$ and $||N||$ for the numerator, and $\sigma^2$ and $D$ for the denominator). There are three considerations regarding cost:
> 1. Computational complexity of estimating from samples: Numerator computations occur in the head dimension and use both K and V caches, making them inherently expensive.
> 2. Sample complexity: This refers to the sample size required for accurate estimation. Theoretically, the sample complexity for the numerator can be higher or lower than that for the denominator, depending on the actual distribution. In practice, we find the sample complexity to be similar. See the table below for estimation errors of the numerator and denominator.
> 3. Memory footprint of samples: Numerator computations require both K and V caches, whereas denominator computations require only K. As a result, the memory footprint for numerator computations is roughly twice that of the denominator for the same sample complexity.
>
>
> |   **niah_multikey_2**  |         |  |   |
> |:------------------:|:-------:|:------------------------------:|:-------------------------------:|
> |  |   |  |         |
> | base sampling rate | ~Tokens | denominator var ( var > 0.001) | numerator trace ( trace > 0.01) |
> | 0.025 |    1000 |4.74% | 2.77% |
> |  0.05 |    2000 |4.45% | 3.16% |
> |   0.1 |    4000 |3.10% | 2.00% |
> |        **qa_1**        |         |      |       |
> | base sampling rate | ~Tokens | denominator var ( var > 0.001) | numerator trace ( trace > 0.01) |
> | 0.025 |     820 |4.91% | 2.67% |
> |  0.05 |    1640 |3.78% | 2.04% |
> |   0.1 |    3280 |2.57% | 1.30% |
> |         **vt**         |         |      |       |
> | base sampling rate | ~Tokens | denominator var ( var > 0.001) | numerator trace ( trace > 0.01) |
> | 0.025 |     820 |5.31% | 2.69% |
> |  0.05 |    1640 |3.63% | 1.72% |
> |   0.1 |    3280 |2.46% | 1.42% |

---

> > ### Comment · Reviewer_5pmM · 2025-11-28
> > **Response for Authors**
> >
> > Thanks for the detailed rebuttal! I keep my rating to maintain the acceptance of this paper.

---

> > > ### Author Response · Authors · 2025-11-28
> > >
> > > We appreciate the reviewer maintaining their opinion of accepting our paper. We have also updated the manuscript to improve the presentation of verified-X algorithms in section 3 (marked in blue). Let us know if there are any more questions.

---

> ### Author Response · Authors · 2025-11-21
> **Response to reviewer 5pmM [2/n]**
>
> ### **Computational Overhead of Budget Calculation /  End-to-end speed evaluation**
> The focus of this paper is long-context decoding. Sparse attention provides little value at shorter sequence lengths, but at very long contexts the KV cache can no longer fit on a single GPU. This forces systems to either offload the KV cache to CPU memory or to shard it across GPUs via context parallelism. In case of offloading, the primary bottleneck is CPU-GPU bandwidth for which we show gains. Evaluating sparse-attention methods under context-parallel execution, requires substantial engineering effort and is beyond the scope of this paper.
>
> ### **Budget partitioning guidance**
> It may be possible to develop a theoretical prescription for budget partitioning that selects optimal top k fractions, potentially yielding a better quality and efficiency trade off. However, because attention score profiles, meaning the cumulative distributions of sorted attention scores, vary significantly across heads and even across queries within the same head (See Figure 2 in the paper), the number of heavy hitters would need to be estimated dynamically. Creating such an adaptive mechanism is an interesting direction for future work.
>
> ### **Early token behavior in small context lengths**
> Sparse attention has little value in small context lengths and should not be applied in small contexts. If the context length for early tokens is small, we should just use full attention.
>
>
> Please let us know if we missed any of the concerns / questions or if there are additional questions. We are happy to clarify futher.

---

### Official Review · Reviewer_VSCh · 2025-10-29

**Soundness:** 3
**Presentation:** 3
**Contribution:** 2
**Rating:** 6
**Confidence:** 3

**Summary:**

This paper introduces vAttention, a verified sparse attention method designed to address the memory and computational bottlenecks of standard scaled dot-product attention (SDPA) in large language models (LLMs) when processing long contexts.

**Strengths:**

*   **Hybrid & Adaptive:** Intelligently combines deterministic "heavy-hitter" tokens (sinks, local, top-k) with stochastic sampling of the tail, adapting per head and query.
*   **State-of-the-Art Performance:** Outperforms strong baselines (e.g., HashAttention, oracle top-p), often matching full-model accuracy with high sparsity (10-15%).
*   **Efficient Long-Context Inference:** Effectively reduces the memory and computational bottleneck of large KV caches, enabling faster generation, especially when the cache is offloaded to CPU.
*   **Practical & Versatile:** Can be integrated with existing approximate top-k methods and works well on diverse, challenging long-context benchmarks.

**Weaknesses:**

*   **Relaxation of Theoretical Guarantees:** The core theoretical contribution provides an \((\epsilon, \delta)\) guarantee for the *entire* attention output. However, the authors explicitly state that in practice, they use a **relaxation that only guarantees the denominator**. While they provide empirical justification (strong correlation with final error), this significantly weakens the formal "verified" claim. The method is no longer provably guaranteed for the final output, but rather for an intermediate component.
*   **Dependence on CLT and Large Sample Assumptions:** The theoretical bounds rely on the Central Limit Theorem (CLT), which holds for "large enough" sample sizes. The paper provides an empirical analysis (Appendix E) showing CLT is tighter than Hoeffding's bound, but the validity of the CLT approximation for *all* layers and heads, especially with small budgets, is not rigorously proven. This makes the "verification" approximate rather than exact.
*   **Circular Dependency in Budget Calculation:** To compute the required sample size \(b\), the algorithm needs to know population statistics like the covariance matrix \(\Sigma\) and the norm \(||N||_2\). Since these are unknown, the method uses a **base sample** (governed by \(f_b\)) to estimate them. The accuracy of the initial estimate directly impacts the final guarantee, creating a potential circularity that is not fully addressed theoretically.
*   **Sensitivity to Hyperparameters:** The method introduces several new hyperparameters (\(f_s, f_l, f_t, f_b, \epsilon, \delta\)). While the paper shows a search can find good values, this adds complexity for practitioners compared to simpler methods like top-\(k\). The "natural configuration" used in AIME2024 is promising but requires validation across more diverse tasks.

* **Computational Overhead of Budget Calculation:** The process of calculating the adaptive budget for each head and query, including drawing a base sample and estimating statistics, introduces **non-trivial overhead**. The paper admits this is done in "naive PyTorch" and that a CUDA kernel is future work. This overhead could offset the speed gains from sparse attention, especially for shorter sequences or GPU-hosted KV caches.
*   **Dependence on Approximate Top-\(k\) Methods:** vAttention's performance is not standalone; it's a framework that incorporates an approximate top-\(k\) method (e.g., HashAttention). The results show that **"more accurate top-\(k\) methods are essential for the overall quality."** Therefore, vAttention's weaknesses are, in part, the weaknesses of its underlying top-\(k\) component. If the top-\(k\) method fails to identify crucial "heavy hitters," vAttention's sampling-based tail approximation may not be sufficient to recover.
*   **No End-to-End Speed Evaluation:** The efficiency claims are primarily supported by a model showing near-linear speedup when the KV cache is on the CPU (Figure 5). There is **no comprehensive evaluation of end-to-end latency or throughput** (tokens/second) comparing vAttention to baselines under equal hardware and sparsity budgets. The gains for GPU-resident KV caches are stated but not demonstrated.

**Questions:**

Please see Weaknesses

---

> ### Author Response · Authors · 2025-11-21
> **Response to VSCh [1/n]**
>
> We thank the reviewer for their time and effort for evaluating our submission
>
> ### **Relaxation of Theoretical Guarantees:**
>
> We realize that our presentation of the denominator-, numerator-, and verified-SDPA (per-head) recipes may have incorrectly suggested that one is a relaxation of another. This was unintended. We will clarify this distinction and upload the revised version by Dec 3.
>
> Denominator, numerator, and per-head attention, the three granularities discussed in the paper, are distinct intermediate computations affected by sparse approximation. For each, we propose a corresponding verified recipe. In general, verified-X means that we can approximately compute X within user-specified $(\epsilon, \delta)$ guarantees. These guarantees are specific to computation X (e.g. denominator, numerator, per-head SDPA etc) and do not directly bound larger encapsulating computations, such as per-layer attention or model output. Thus verified-denominator, verified-numerator and verified-SDPA (per-head) are independent algorithms guaranteeing specific computations, not relaxations of one another.
>
> Given these three options, the selection of X depends on two factors
> - Control: How interpretable and fine-grained the control is that verified-X provides over the quality of larger computation encapsulating X. We use per-layer attention error as a larger computation.
> - Cost: How expensive the corresponding budget computation is under each verified-X option
>
> As shown in Figure 1 (right), and in the newly added more extensive plots for both numerator and denominator in Appendix F (Figures 16 and 17), numerator and denominator offer similarly fine-grained control over model errors. The correlation between the layer-level attention output error (i.e., the error of the attention module after combining all heads and applying the O projection) and the user-specified $\epsilon$ remains very high for both. Thus, both are strong candidates. We prefer verified-denominator because it is simpler and requires access only to the base sample of keys, whereas verified-numerator and verified per-head attention require access to both keys and values, effectively doubling the memory needed for the base sample.
>
>
>
> ### **Verification in small-budget regimes and CLT**
>
> It is true that CLT cannot be applied at small budgets. However, the target regime of the problem is a long context where reasonable budgets also provide us with a large sample size. For example at 128K or 1M context size (the real application of sparse attention) will give us sample sizes of 12K or 100K at 10% sparsity.  Such large sample sizes are good enough for application of CLT. We can see this in practice ( See QQ plots for the estimator in newly added Appendix H) and also in theory. For example, In case of denominator estimation, all the moments of the estimator are bounded (specifically the 3rd moment). Thus, we can apply Berry-Esseen Theorem to know that the distribution converges at the rate of $1/\sqrt{n}$.
>
> ### **Circular Dependency in Budget Calculation**
>
> Estimating unknown statistics and using them is a standard procedure in developing statistical algorithms. This is popularly referred to as plug-in estimators. Some references that talk about this are “All of Statistics, Wasserman (Chapter 7)”, “An Introduction to Bootstrap” , Efron and Tibshirani.
> To show how well we approximate meaningful statistics with the sample sizes that we choose, we show the results below. We find that even with small samples we are able to approximate the statistics with very small errors. This is expected since these estimators have fast convergence.
>
> |   **niah_multikey_2**  |         |  |   |
> |:------------------:|:-------:|:------------------------------:|:-------------------------------:|
> |  |   |  |         |
> | base sampling rate | ~Tokens | denominator var ( var > 0.001) | numerator trace ( trace > 0.01) |
> | 0.025 |    1000 |4.74% | 2.77% |
> |  0.05 |    2000 |4.45% | 3.16% |
> |   0.1 |    4000 |3.10% | 2.00% |
> |        **qa_1**        |         |      |       |
> | base sampling rate | ~Tokens | denominator var ( var > 0.001) | numerator trace ( trace > 0.01) |
> | 0.025 |     820 |4.91% | 2.67% |
> |  0.05 |    1640 |3.78% | 2.04% |
> |   0.1 |    3280 |2.57% | 1.30% |
> |         **vt**         |         |      |       |
> | base sampling rate | ~Tokens | denominator var ( var > 0.001) | numerator trace ( trace > 0.01) |
> | 0.025 |     820 |5.31% | 2.69% |
> |  0.05 |    1640 |3.63% | 1.72% |
> |   0.1 |    3280 |2.46% | 1.42% |

---

> ### Author Response · Authors · 2025-11-21
> **Response to VSCh [2/n]**
>
> ### **Sensitivity to Hyperparameters:**
>
> Most sparse attention methods, including top-k approaches need some calibration of sorts. Choosing the hyper parameters can be thought of as a calibration step, similar to how we need to create a model specific channel config in Double Sparsity or MLP weights need to be trained for HashAttention. Additionally, we provide easy to use scripts where we can quickly search for the parameters in the specified grid to simplify the process.
>
> Moreover, there is a natural and stable choice for most parameters. In our experiments, we generally use fixed values for $f_s$​ and $f_l$​, and perform a coarse-grained search for the remaining parameters. To demonstrate stability, we conducted an extensive sensitivity analysis, now presented in Appendix I. This analysis confirms our expectation that choosing safe values for $f_s​, f_l​, f_t​, f_b$​ (for example, $f_s >= 2 , f_l >=64, f_t, f_b >= 0.025$ ) places the method on the standard quality–efficiency tradeoff curve, which can then be traversed through appropriate choices of $\epsilon,\delta$.
>
> Natural configurations can be used across the board (specifically for in-the-wild deployment). For instance if we use the same parameters as we used for AIME ($f_s = 128 , f_l =128, f_t, f_b = 0.025, \epsilon=\delta=0.05$) on RULER-HARD, we get the following performance which essentially matches the full attention performance at 10-15% tokens.
>
>
> | Dataset | Density | Quality | Full Attention (reference) |
> | :--- | :--- | :--- | :--- |
> | qa_1 | 0.143 | 80.5 | 80.5 |
> | qa_2 | 0.136 | 51 | 51.5 |
> | niah_multikey_2 | 0.111 | 99.5 | 99.5 |
> | niah_multikey_3 | 0.114 | 100 | 100 |
> | vt | 0.16 | 97.4 | 97.4 |
> | fwe | 0.103 | 93.17 | 93.17 |
>
>
>
>
> ### **Computational Overhead of Budget Calculation /  End-to-end speed evaluation:**
>
> The focus of this paper is long-context decoding. Sparse attention provides little value at shorter sequence lengths, but at very long contexts the KV cache can no longer fit on a single GPU. This forces systems to either offload the KV cache to CPU memory or to shard it across GPUs via context parallelism. In case of offloading, the primary bottleneck is CPU-GPU bandwidth for which we show gains. Evaluating sparse-attention methods under context-parallel execution, and comparing highly optimized implementations across such distributed setups, requires substantial engineering effort and is beyond the scope of this paper.
>
> ### **Dependence on Approximate Top-(k) Methods:**
>
> It is true that vAttention quality depends on the quality of underlying top-k method. However, the contribution of vAttention is orthogonal. vAttention tackles the Achilles’ heel of top-k sparse attention. As shown in our work and prior studies (e.g. MagicPig), even oracle top-k cannot reach full-attention quality, and approximate methods perform even worse. By filling this crucial gap between oracle top-k and full attention, vAttention reopens the door for approximate top-k research—showing that together with vAttention, improving top-k prediction accuracy can meaningfully close the gap to full attention.
>
>
> Please let us know if we missed any of the concerns / questions or if there are additional questions. We are happy to clarify further.

---

> > ### Comment · Reviewer_VSCh · 2025-11-27
> >
> > Thanks for your reply, which is very impressive. I would therefore enhance my confidence from 3 to 4.

---

> > > ### Author Response · Authors · 2025-11-28
> > > **Response to reviewer**
> > >
> > > We appreciate the reviewer updating their confidence in accepting our paper. We have also updated the manuscript to improve the presentation of verified-X algorithms in section 3 (marked in blue).

---

### Author Response · Authors · 2025-11-21
**Response to all reviewers (summary of changes) [1/n]**

We thank all the reviewers for their time and efforts to evaluate our manuscript. Here we summarize the main points, rebuttal and additional evidence added to the paper.

We appreciate the reviewers’ high ratings for our paper in soundness (3), presentation (3), and contribution (2.5). Across reviewers, several strengths were consistently highlighted. They emphasized the paper’s clear hybrid design that combines deterministic heavy hitters with stochastic sampling, its principled foundations with user-controllable error guarantees, and its strong empirical performance that often matches full attention at high sparsity. Reviewers also praised the clarity of the writing, the comprehensive experiments and ablations, and the practical relevance and compatibility with existing top-k methods for efficient long-context inference.
The main concerns raised by the reviewers were as follows. We summarize our response. Other concerns are addressed in the individual responses.

| Concern | Raised by   | Summary of response |
|--------------------------------------------------------------------------------------------------|-------------|--------------------------------------------------------------------------------------------------------------------------------------------------------------------------------------------------------------------------------------------------------------------------------------------------------------------------------------------------------------------------------------------------------------------------------------------------------------------------------------------------------------------------------------------------------------------------------------------------------------------------------------------------------------------------------------------------------------------------------------------------------|
| By using denominator-verified recipes, we have relaxed the per-head-attention-verified guarantee | R1,R2,R3,R4 | We realize that our current presentation was not sufficiently clear, which contributes to the confusion. To clarify, Denominator-verified, numerator-verified, and per-attention-head-verified are three distinct, self-contained algorithms. Each provides verified approximations for specific computations: the denominator, the numerator, and per-head attention, respectively. We have included additional results to demonstrate the performance of each method and to provide the rationale for our ultimate selection of the denominator-verified algorithm.                                  |
| Is use of CLT justified    | R1, R4      | We include QQ plots showing that the observed distribution is indeed close to normal.                                                                                     |
| Can we use plug-in estimations in our theorems in practice?      | R1, R4      | The substitution of consistent estimators into theoretical bounds is a standard practice in statistical learning theory (the 'plug-in principle'). To ensure this holds in our specific setting, we empirically validated the quality of our estimators across multiple datasets and sample sizes. As shown in newly added Table 11 (Appendix) , the convergence of these estimators is rapid, demonstrating that the theoretical guarantees translate effectively to practice.                                                 |
| Novelty of vAttention . Is [1] the same as vAttention?        | R4          | While the idea of combining top-k selection with sampling in a form related to vAttention is briefly mentioned in [1] (and we thank the reviewer for pointing this out), the paper is primarily theoretical and focuses on a different estimation scheme aimed at proposing a sub-quadratic attention mechanism for training from scratch. It does not translate that formulation into a practical, usable algorithm for inference—the central goal of vAttention—and presents only two small-scale toy training experiments for its approximation. In contrast, vAttention develops this idea into a practical inference-time optimization method and provides a rigorous empirical evaluation on state-of-the-art, off-the-shelf open-source models. |
| Speed up evaluation with custom CUDA kernels missing                                             | R1,R2,R3    | We focus our evaluation on the offloading scenario, where we already demonstrate significant speedups. While we agree that custom CUDA kernels could further enhance performance, implementing a context-parallel sparse attention kernel is a complex systems challenge distinct from the algorithmic contributions of this paper. We treat kernel-level optimization as orthogonal work that would strictly improve upon the baselines we have established.                                                           |

[1] Haris, Themistoklis. "kNN Attention Demystified: A Theoretical Exploration for Scalable Transformers." The Thirteenth International Conference on Learning Representations

---

> ### Author Response · Authors · 2025-11-21
> **Response to all reviewers (summary of changes) [2/n]**
>
> In response to reviewer comments, we have performed and included the following new experiments in the appendix of the paper. Section headers for new content are marked in blue:
>
> **Appendix F:** Comprehensive ablation of the effect of $\epsilon$ and $\delta$ choices on sparsity and attention layer output errors for both denominator-verified and numerator-verified approaches. The results show that both approaches provide equally effective fine-grained control over layer errors for the end user.
>
> **Appendix G:** Evaluation of the quality of estimations for the statistics required in budget computation for both numerator and denominator. The results demonstrate that even with small base samples, the estimation quality is very high.
>
> **Appendix H:** Validation of the Central Limit Theorem (CLT). We show that the estimators follow a distribution close to normal through QQ plots generated from actual attention data.
>
> **Appendix I:** Sensitivity analysis for different parameters in vAttention. The results indicate that safe values can be chosen for $f_s$, $f_l$, $f_t$, and $f_b$, and that $\epsilon$ and $\delta$ can be used to achieve the desired sparsity.
>
> **Appendix J:** we have extended the empirical evaluation to include 5 SOTA models (1B sized to 30B sized) including dense and moe architectures, and compared against a total of 6 SOTA sparse attention baselines (DoubleSparsity, HashAttention, MagicPig, OracleTopK,OracleTopP,PQCache,Quest)
>
> In addition to these experiments, further data is provided in the individual responses.
>
> We believe we have adequately addressed the concerns raised by the reviewers and are happy to provide further clarifications if needed. If the reviewers feel their concerns have been resolved, we kindly request that they consider revising their scores accordingly.

---

### Meta-Review · Area_Chair_ffFr · 2026-01-08

**Summary:**

This paper combines top-k attention with a token sampling approach, seeking an interpolation of the two. It seeks provable and tunable approximation guarantees of the attention function and a practical method that can be deployed to save time and memory without sacrificing model quality. Numerous experiments are performed against many benchmarks to validate the method's performance.

This paper proposes a clear hybrid design that combines deterministic heavy hitters with stochastic sampling, its principled foundations with user-controllable error guarantees, and its strong empirical performance that often matches full attention at high sparsity. Although the idea of combining sampling and top-k attention is already mentioned in prior research, this paper clearly showcases the practicality of such an approach and performs rigorous implementable algorithm, which shows its novelty. The presentation and theoretical completeness concerns were mentioned by the reviewers, however, after the careful rebuttal and revision, I think the authors solve most of them. The only remaining issue is that this vAttention method does not shown an end-to-end gain in real implementation in terms of speed and efficiency, as the authors argue that this needs custom CUDA kernel development, which weakens its significance, but I think it's tolerable.

**Reviewer Concerns:**

The concerns regarding the novelty, presentation and theoretical completeness have been addressed by the rebuttal, while the concern on the speed evaluation with CUDA kernel is still outstanding.

**Reviewer Scores:**

The original scores are 6(5pmM), 2(rduG), 6(YGRV), 6(VSCh). I would expect a change to 6,6,6,6 after a full rebuttal.

---

### Decision · Program_Chairs · 2026-01-26

Accept (Poster)